# Phylogenetic Analysis of the Genus *Planaphrodes* Hamilton (Hemiptera, Cicadellidae, Aphrodinae) Based on Morphological Characteristics, with Revision of Species from China, Korea and Japan

**DOI:** 10.3390/insects14030291

**Published:** 2023-03-16

**Authors:** Zonglei Liang, Jin-Hyung Kwon, Masami Hayashi, Christopher H. Dietrich, Wu Dai

**Affiliations:** 1Key Laboratory of Plant Protection Resources and Pest Management of Ministry of Education, Entomological Museum, Northwest A&F University, Yangling 712100, China; 2School of Applied Biosciences, College of Agriculture & Life Sciences, Kyungpook National University, Daegu 702-701, Republic of Korea; 3Kyushu University Museum, Fukuoka 812-8581, Japan; 4Illinois Natural History Survey, University of Illinois, Champaign, IL 61820, USA

**Keywords:** leafhopper, phylogeny, morphology, new species, new synonyms, taxonomy

## Abstract

**Simple Summary:**

*Planaphrodes* is a genus of Aprhodinae distributed in the Palaearctic region from Portugal to Japan. This paper reconstructs phylogenetic relationships among species of *Planaphrodes* based on morphological characteristics for the first time, elucidates the phylogenetic status of the genus and describes two new species found in China.

**Abstract:**

A morphology-based phylogeny of the Holarctic leafhopper genus *Planaphrodes* Hamilton is reconstructed for the first time based on 39 discrete male adult morphological characters. The results support the monophyly of *Planaphrodes*, with the included species forming two monophyletic lineages defined mainly by the number and location of aedeagus processes. The position of *Planaphrodes* in the Aphrodini was resolved as follows: (*Stroggylocephalus* + (*Anoscopus* + (*Planaphrodes* + *Aphrodes*))). The fauna of *Planaphrodes* from China, Japan and Korea are reviewed and six species are recognized, including two new species: *P. bifasciatus* (Linnaeus), *P. sahlbergii* (Signoret), *P. nigricans* (Matsumura), *P. laevus* (Rey), *P. baoxingensis* sp. nov. (China: Sichuan) and *P. faciems* sp. nov. (China: Hubei). *Acocephalus alboguttatus* Kato, 1933 syn. nov. and *Aphrodes daiwenicus* Kuoh, 1981 syn. nov. are considered junior synonyms of *Planaphrodes sahlbergii* (Signoret, 1879). *Planaphrodes bella* Choe, 1981 is a junior synonym of *Planaphrodes nigricans* (Matsumura, 1912). A checklist and key to species of *Planaphrodes* are provided.

## 1. Introduction

Aphrodini (subfamily Aphrodinae) is a small group of leafhoppers comprising four genera and fifty-eight species in the world, distributed primarily in the Palaearctic realm [1]. Leafhoppers of the tribe are common on herbaceous plants, usually in meadows and pastures, and some species live and feed on roots beneath the surface litter [2]. Some species also breed on leguminous crops, e.g., alfalfa [3,4]. Two species, *Anoscopus albifrons* (Linnaeus) and *Aphrodes bicincta* Schrank, are known vectors of phytoplasma plant pathogens [5,6]. Unfortunately, little is known about the ecology of most species of the group, partly due to their cryptic lifestyles, which make them seldom encountered in routine collecting [7]. Species of Aphrodini may be recognized by their dorsally flattened produced heads with ocelli on the anterior margin distant from the eyes and lateral frontal sutures extended ventromesad of the ocelli and by the basally narrow male subgenital plates and slender, sinuate style [7,8,9]. The genera are distinguished by differences in head morphology and proportions and the shape and distribution of spines on the aedeagus [7].

In Hamilton’s comprehensive review of the Northern Hemisphere Aphrodina, he erected *Planaphrodes* with the type species *Acucephalus tricincta* (Curtis) and included fifteen species based on his examination of a series of type material for species in the genus. Species of *Planaphrodes* are widely distributed in the Palearctic region. Due to low population densities and dull appearance, this genus is not well represented in collections and remains poorly studied. *Planaphrodes*, as defined by Hamilton [7], differs from other Aphrodini in having a relatively long horizontal crown and a lamellate aedeagal shaft. Although Hamilton [7] provided some notes and presented an intuitive diagram of evolutionary relationships between species based on the structure of the aedeagus, he did not redescribe the known species or provide a key for their identification. Moreover, *Acucephalus alboguttatus* Kato was excluded from this genus for lack of male genitalia data, and *P. grisea* Mitjaev was included but has the styles fused to the aedeagus and is probably an abnormal form of *P. laeva* Rey. Emeljanov [10] indicated that *P. sahlbergi* sensu. Hamilton [7] was a misidentification of *P. nigricans* and also treated *Acocephalus bifasciatus* var. *guttatus* Matsumura, 1912 [11], *Aphrodes japonicus* Dlabola, 1960 [12] and *Aphrodes mongolicus* Dlabola, 1965 [13] as junior synonyms of *P. sahlbergi* in his treatment of Mongolian fauna. Anufriev [14], synonymized *P. mongolicus* (Dlabola, 1965) [13] with *P. guttatus* (Matsumura, 1912) [11]. Based on observation of Russian specimens, Anufriev [15] pointed out that *P. nigricans* was valid and *P. guttatus* should be synonymized with *P. sahlbergi* because *P. nigricans* and *P. guttatus* had been confused by Hamilton [7] when he checked Matsumura’s type specimens. Later, Choe [16] added the species *P. bella* from Korea. Logvinenko [17] described another species, *Aphrodes* (*Planaphrodes*) *nisamiana* from Russia. Recently, *Planaphrodes dobrogicus* (Cantoreanu, 1968) [18] was placed in synonymy under *Planaphrodes angulaticeps* (Emeljanov, 1964) [19] by Gnezdilov [20], and *Planaphrodes* has never been revised comprehensively. Species of this genus can exhibit considerable variation in color pattern, particularly between males and females; thus, species have been traditionally defined based on the structure of the male genitalia, particularly the shape and arrangement of spines of the aedeagus, which appears to be more stable. Tishechkin [21] compared the male vibrational calling signals and genitalia of four species of Central Asian and Western Palearctic *Planaphrodes*, noting that some morphologically distinct species with broadly overlapping distributions have similar courtship calls but occupy different plant communities.

Information on Chinese *Planaphrodes* is scattered across the literature. Oshanin [22] first recorded *P. bifasciatus* (as *Acocephalus bifasciatus*) from Western China. Kato [23] and Jacobi [24] reported *P. alboguttatus* (as *Acocephalus alboguttatus*) and *P. sahlbergii* (as *Aphrodes sahlbergi*) from China (Manchuria), respectively. There are only two species (*P. nigricans* and *P. guttatus*) recorded from Japan [25] and four species from Korea (*P. alboguttata*, *P. bella*, *P. nigricans* and *P. sahlbergi*) [26]. However, to our knowledge, no study has yet extensively explored the geographical diversity of *Planaphrodes*, especially in China, Japan and Korea.

In this study, we use comparative morphological data to reconstruct phylogenetic relationships among species of *Planaphrodes*, elucidate the phylogenetic status of the genus and clarify its relationship to other genera of Aphrodini.

## 2. Materials and Methods

### 2.1. Taxa and Terminology

Specimens examined in this study are deposited in the Entomological Museum, Northwest A&F University, Yangling, Shaanxi, China (NWAFU), Institute for Agro-Environmental Sciences, National Agriculture and Food Research Organization, Tsukuba, Japan (NIAES), Kyungpook National University, Daegu, South Korea (KNU), Kyushu University Museum, Fukuoka, Japan (ELKU), Nankai University, Tianjin, China (NKU), the Systematic Entomology Collection, Hokkaido University, Sapporo, Japan (SEHU) and the University Museum of the University of Tokyo, Tokyo, Japan (UMUT). The male abdomens of the specimens were removed and cleared in 10% NaOH solution for 48 h, rinsed in water and then suspended in glycerin for further dissection and examination. After examination, each was moved to fresh glycerine and stored in a micro vial pinned below the specimen.

All specimens were examined with a Leica ZOOM 2000 stereomicroscope. The habitus images of adults, including forewings and hind wings, were captured with a ZEISS SteREO Discovery V20 Stereoscopic microscope equipped with ZEN cartographic software. The male genitalia were photographed using a Nikon Y-IDT stereomicroscope with Auto-Montage cartographic software, then adjusted by Photoshop CS6.

Morphological terminology follows Hamilton [7] and Dietrich [27]. Body length was measured in mm from the apex of the head to the apex of the forewings.

### 2.2. Taxon Sampling and Morphological Characters

The phylogenetic analysis included all 16 recognized species of *Planaphrodes* and 7 outgroup taxa. Because specimens were not available for all species, 6 ingroup and 7 outgroup species distributed in China were examined and morphological characters of other species were scored based on illustrations and descriptions in the literature [4,17,18,28,29,30,31,32]. Representatives of the other recognized genera of Aphrodini and trees were rooted to the outgroup, *Xestocephalus* (Xestocephalini), based on the phylogenetic results of Dietrich et al. [33] and Skinner et al. [34] Morphological data consisted of 39 discrete binary and multistate characters treated as unordered and of equal weight from male specimens. Inapplicable characters were indicated as ‘-’ and unobserved states with ‘?’. Characters and states are listed below and the matrix is in Table 1. Comparative morphology of Aedeagus and style of *Planaphrodes* is provided in Table 2.

List of morphological characters used in phylogenetic analyses, body and head (Figures 4–16):
0.Microsculpture of crown and pronotum: 0, glabrous (Liang et al. 2021 [35], Figure 1A,D); 2, finely striate; 1, shagreened (Figures 4A–G,I and 5A–D).1.Width of head: 0, wider than pronotum (Figures 4A–G,I and 5A–D); 1, narrower than pronotum.2.Crown: 0, blunt, almost parallel-margined; 1, slightly produced (Liang et al. 2021 [35], Figure 1A,D); 2, strongly produced or elongate (Figures 4A–G,I and 5A–D); (State: 0, strongly narrower than midwidth of pronotum; 1, same as midwidth of pronotum; 2, significantly wider than midwidth of pronotum).3.Vertex: 0, crown rounded to face, transition poorly delimited (Liang et al. 2021 [35], Figure 1B,E); 1, with anterior margin strongly carinate, transition from crown to face well delimited (Figures 4H and 5E–D).4.Crown anterior margin: 0, unicolorous or without spots (Liang et al. 2021 [35], Figure 1A,D); 1, with numerous small bright spots (Figures 4A–G,I and 5A–D).5.Crown: 0, slightly convex, smooth (Liang et al. 2021 [35], Figure 1A,D); 1, flat, with distinct medial carina and two slightly elevated keels behind the ocelli (Figures 4A–G,I and 5A–D).6.Wings: 0, transparent, membranous (Liang et al. 2021 [35], Figure 1B,E); 1, opaque, leathery (Figures 4H, 5E–H and 11A).7.Venation: 0, not elevated (Liang et al. 2021 [35], Figure 1B,E); 1, elevated (Figures 4H and 5E–H).
Male genitalia:
8.Genital capsule: 0, cylindrical (Liang et al. 2021 [35], Figure 2A); 1, conical (Figures 6D, 9B,C, 14B,15B and 16B).9.Lobes of the pygofer posterior margin: 0, significantly produced, rounded (Liang et al. 2021 [35], Figure 2A); 1, absent; 2, folded into cavity, forming a partially sclerotized barrier, like an interconnecting membrane in some leafhoppers, with strigate sculpture (Figures 6A,B, 9A, 14A, 15A and 16A).10.Pygofer lobe posterior margin: 0, with papillae or microsetae (Liang et al. 2021 [35], Figure 2A); 1, without papillae or microsetae (Figures 6A,B, 9A, 14A, 15A and 16A).11.Setae of pygofer: 0, 2 rows of large macrosetae near the base of lobe (Liang et al. 2021 [35], Figure 2A); 1, macrosetae in irregular tuft surrounding anal tube; 2, microsetae, scattered (Figure 14A,B).12.Pygofer appendage: 0, inner, a small ventrad angular projection (Liang et al. 2021 [35], Figure 2B); 1, outer, posteroventrally directed harpoon-shaped process; 2, a curved dorsal tarpering process; 3, outer, a posteriorly directed swollen and curved dorsal hook–shaped process (Figure 6C).13.Valve shape: 0, approximately rectangular (Liang et al. 2021 [35], Figure 2C); 1, expanded sickle shaped; 2, depressed trapezoidal (Figures 6E, 9D, 14C, 15C and 16C).14.Subgenital plate shape: 0, broad with apex rounded (Liang et al. 2021 [35], Figure 2C); 1, ligulate with apex attenuate (Figures 6E, 9D, 14C, 15C and 16C).15.Setae on subgenital plate: 0, two rows of large submarginal setae and several rows of hairlike setae mesally (Liang et al. 2021 [35], Figure 2C); 1, long setae, submarginal and mesal; 2, short setae, submarginal and at apex; 3, microsetae, scattered (Figures 9D and 14C).16.Style shape: 0, S-shaped (Liang et al. 2021 [35], Figure 2D); 1, crescent-shaped, slenderer, significantly bent; 2, crescent-shaped, not bent (Figures 9E,F and 14D); 3, crescent-shaped, broader, significantly bent (Figures 6F,G, 15D and 16D).17.Style apophysis apex: 0, acuminate, with foot-like extension (Liang et al. 2021 [35], Figure 2D); 1, blunt, slightly expanded then tapering; 2, blunt, uniform width; 3, blunt, strongly expanded; 4, obliquely truncate, broadened (Figures 9E,F and 14D); 5, blunt, strongly bent and expanded (Figures 6F,G, 15D and 16D).18.Style apophysis ventral margin: 0, without denticles or areoles (Liang et al. 2021 [35], Figure 2D); 1, with denticles, areoles absent; 2, with denticles and areolate submargin (Figures 6F,G, 9E,F, 14D, 15D and 16D).19.Connective: 0, Y-shaped with a short median anterior lobe (Liang et al. 2021 [35], Figure 2D); 1, Y-shaped with posterior stem divided; 2, Y-shaped without median anterior lobe (Figures 6F,G, 9E,F, 14D, 15D and 16D).20.Aedeagus dorsal apodeme: 0, significantly longer than 1/2 length of aedeagal shaft (Liang et al. 2021 [35], Figure 2F,H); 1, almost 1/2 length of aedeagal shaft (Figures 7A–F, 9G,H, 14F, 15F and 16F).21.Aedeagal shaft: 0, cylindrical (Liang et al. 2021 [35], Figure 2E,G); 1, flattened laterally (Figures 7G,H, 9I,J, 14E, 15E and 16E).22.Aedeagal shaft dorsal lamella: 0, present (Liang et al. 2021 [35], Figure 2F,H); 1, absent (Figures 7A–F, 9G,H, 14F, 15F and 16F).23.Shape of aedeagal shaft in lateral view: 0, straight, evenly broad (Figure 9G,H); 1, slightly curved, thin, atrium widened, shaft tapering; 2, arcuate, base evenly broad, apical 1/2 gradually tapering; 3, straight, uniformly slender; 4, straight, thin, slightly broader in middle than at both ends (Table 2: Aedeagus lateral view of *P. elongatus*, *P. iranicus* and *P. nisamiana*); 5, straight, broad, apical 1/3 abruptly tapered (Figure 14F); 6, slightly curved, strongly widened in middle 1/3 (Figures 7A–F, 15F and 16F).24.Aedeagal shaft apical denticles: 0, absent (Figures 7G,H, 9I,J, 14E, 15E and 16E); 1, present (Table 2: Aedeagus caudal view of *P. bifasciatus*).25.Pairs of processes on aedeagal shaft: 0, 0 (Liang et al. 2021 [35], Figure 2E–H); 1, 1; 2, 2 (Table 2: Aedeagus lateral view of *P. vallicola*); 3, 3 (Figures 9G,H and 14F); 4, 4 (Figures 7A–F and 15F); 5, 5 (Figure 16F).26.Apical spines of aedeagus: 0, long, thin petal-shaped; 1, long, broad petal-shaped; 2, short spine; 3, tiny hook-like (Figures 7A–F, 9G,H, 14F, 15F and 16F); 4, widened hook (Table 2: Aedeagus lateral view of *P. angulaticeps* and *P. iranicus*).27.Apical spines of aedeagus, position: 0, arising laterally (Figure 14E,F); 1, arising caudally (Figures 7A–H, 9G–J, 15E,F and 16E,F).28.Apical spines of aedeagus, orientation: 0, directed posteroventrad (Figures 7A–H, 9G–J, 15E,F and 16E,F); 1, directed ventrad; 2, directed anteroventrad (Figure 14E,F).29.Lateral processes of aedeagus: 0, short spine (Figures 9G,H and 14F); 1, large crescent-shaped process; 2, spinule (Figures 7A–F, 15F and 16F); 3, triangle (Table 2: Aedeagus lateral view of *P. nisamiana*).30.Lateral processes of aedeagus, orientation: 0, divergent laterally, anteroventrad (Table 2: Aedeagus lateral and caudal view of *P. bifasciatus*, *P. monticola*, *P. vallicola*); 1, directed anteroventrad (Figures 7A–H, 9G–J, 14E,F, 15E,F and 16E,F); 2, posteroventrad.31.Caudal processes of aedeagus: 0, absent (Table 2: Aedeagus lateral view of *P. vallicola*); 1, slender spines (Table 2: Aedeagus lateral view of *P. bifasciatus*, *P. elongatus*, *P. iranicus*, *P. nisamiana*); 2, poorly developed tooth (Table 2: Aedeagus lateral view of *P. mondicus*); 3, tiny tooth (Figures 7A–F, 15F and 16F); 4, large shark fin (Figures 9G,H and 14F).32.Bases of caudal processes of aedeagus: 0, separated; 1, connected (Figures 7G,H, 9I,J, 15E and 16E); 2, fused for most of length (Figure 14E Table 2: Aedeagus caudal view of *P. lusitanicus*).33.Relative distance between lateral and caudal processes of aedeagus: 0, lateral processes much higher than caudal processes (Table 2: Aedeagus lateral view of *P. angulaticeps*, *P. araxicus*, *P. modicus* and *P. monticola*); 1, lateral processes slightly higher than caudal processes (Figure 14F); 2, lateral processes and caudal processes at the same level (Figure 9G,H); 3, lateral processes lower than caudal processes Figures 7A–F, 15F and 16F) (0, processes with bases well separated, greater than the lateral processes length; 1, processes with bases close to each other, tips of lateral processes not reaching base of caudal processes).34.Dorsal processes of aedeagus: 0, absent (Figures 9G,H and 14F); 1, acute triangle (Figure 15F); 2, obtuse triangle (Figure 16F); 3, tooth (Table 2: Aedeagus lateral view of *P. bifasciatus*, *P. monticola* and *P. nigritus*); 4, long and strongly divergent process (Figure 7A–F and Table 2: Aedeagus lateral view of *P. vallicola*).35.Dorsal processes of aedeagus orientation: 0, directed posteroventrad (Figures 15E,F and 16E,F); 1, directed downward (Table 2: Aedeagus lateral and caudal view of *P. bifasciatus*, *P. monticola*, *P. nigritus*); 2, directed dorsad (Figure 7A–F and Table 2: Aedeagus lateral and caudal view of *P. vallicola*);36.Relative positions of dorsal and lateral processes of aedeagus: 0, dorsal processes higher than lateral processes (Figures 7A–F, 9G,H, 15E,F and 16E,F); 1, dorsal processes lower than lateral processes (Table 2: Aedeagus lateral view of *P. monticola*).37.Shape of gonopore: 0, circle (Liang et al. 2021 [35], Figure 2E,G); 1, willow leaf shaped; 2, flat ellipse; 3, teardrop-shaped (Figures 7G,H, 9I,J, 14E,F, 15E,F and 16E,F).38.Position of gonopore: 0, apical (Liang et al. 2021 [35], Figure 2E,G); 1, proximal 1/4; 2, proximal 1/3 (Table 2: Aedeagus caudal view of *P. vallicola*); 3, middle (proximal 1/2) (Table 2: Aedeagus caudal view of *P.iranicus* and *P. nisamiana*); 4, proximal 3/4 (Figures 7G,H, 9I,J, 14E, 15E and 16E).

### 2.3. Phylogenetic Analysis

The data matrix was analyzed using TNT1.1 [37] initially using the traditional search method. The equal weighting analysis with 1000 replicates (trees to save = 10, random seed = 0) was used to produce the final phylogenetic estimate, and character changes were mapped using WinClada 1.0 [38]. The branch support was calculated by using TNT to search for suboptimal trees and calculating decay indices (Bremer support) as the difference in length between the original MP trees and the shortest trees incompatible with each resolved branch. Bootstrap replicates [39] were analyzed to assess node support with 1000 replicates and 50% was used as the cut off value. The illustrated cladograms were edited using Adobe illustrator CS 6.0. and Adobe Photoshop CS6.0.

Ancestral area probabilities were estimated in RASP v.4.0 [40] using the Bayesian binary MCMC (BBM) method and default parameter settings. The tree from Figure 1 inferred from the morphological matrix by TNT was used to infer the ancestral distribution of each clade. The following biogeographic regions were assigned to extant taxa based on known distributions: (A) Europe, (B) Middle East and Western Asia, (C) Central Asia (including western China), (D) Eastern Asia (including central and eastern China, Korea and Japan). Taxa that occur in multiple areas were assigned multiple states.

## 3. Results

### 3.1. Phylogeny

Maximum parsimony analysis yielded four most parsimonious trees of length = 110, consistency index (CI) = 0.79 and retention index (RI) = 0.82. To simplify the presentation of results, only one of the MP trees is shown, with decay indices and boostrap support indicated for branches in Figure 1. Character state changes are indicated in Figure 2. All MP trees are consistent with the previous genus-level classification, with all genera of Aphrodini recovered as monophyletic based on the included species. Three branches with no support values indicated in Figure 1 collapsed in the strict consensus of all MP trees. Our analysis indicates that *Aphrodes* and *Planaphrodes* are sister groups (bb = 77, Br = 3), supported by the following synapomophies: crown and pronotum finely striate (char. 0: 1), crown flat, with distinct medial carina and two slightly rising keels behind the ocelli (char. 5: 1), forewings opaque and leathery (char. 6: 1), valve depressed and trapezoidal (char. 13: 2), styles crescent-shaped (char. 16: 0) and style apophysis obliquely truncate and broadened (char. 17: 4). The positions of *Stroggylocephalus* and *Anoscopus* relative to each other were not consistently resolved by our analysis. Broader analyses of this group including a larger taxon sample and more distantly related outgroups are needed to fully resolve the phylogenetic status of this tribe.

*Planaphrodes* received relatively strong branch support (bootsrap, bb = 83; Bremer, Br = 3) and is supported by the following synapomorphies: crown strongly produced (char. 2: 2), crown anterior margin with numerous small bright spots (char. 4: 1), aedeagal shaft significantly flattened laterally (char. 21: 1), aedeagal shaft partial widened in lateral view: (char. 23: 4,5,6), teardrop-shaped gonopore: (char. 37: 3) and gonopore situated at shaft proximal 1/3 (char. 38: 2). Among these synapomorphies, the flattened aedeagal shaft is unique to *Planaphrodes* species, which corroborates the monophyly of the genus. Relationships among *Planaphrodes* species are also stable and all but one species of the genus grouped into two sister clades.

*Planaphrodes vallicola* is sister to the remaining *Planaphrodes* (bb = 83; Br = 3). The other *Planaphrodes* species grouped into two clades. *P. vallicola* is unique in having the posterior pair of aedeagal processes situated more dorsally than in other species of the genus, in this respect more closely resembling species of *Aphrodes*. Nevertheless, *P. vallicola* shares the compressed aedeagal shaft present in other *Planaphrodes*, supporting its retention in this genus.

The clade comprising the rest of the species of *Planaphrodes* isunited by the presence of more than four processes on the aedeagal shaft (char. 25: 3) and the position of the gonopore near the base of the shaft (char. 38: 4).

Clade 1 comprises six species. *Planaphrodes bifasciatus* is separated based on aedeagal shaft with apical denticles (char. 24: 1) and lateral processes and caudal processes almost at the same level (char: 33: 2). The sub-branch is supported by aedeagal shaft with more than four pairs of processes (char. 25: 4) and tiny tooth caudal processes (char. 31: 3). These species also have the aedeagal shaft distinctly broadened in lateral view. *P. nigricans*, *P. baoxingensis* and *P. faciems* are supported as a monophyletic group by the presence of tiny lateral processes (char. 29: 2) and lateral processes lower than caudal processes (char. 33: 3). The triangular dorsal processes (char. 34: 2) and ventral orientation of the dorsal aedeagal processes (char. 35: 0) are synapomorphic for the *P. baoxingensis* and *P. faciems* sister pair.

Clade 2 species are united by synapomorphies: lateral processes directed anteroventrad (char. 30: 1) and aedeagal shaft with three pairs of processes and without dorsal processes (char. 34: 0). Although the tree in Figure 1 placed nine species in Clade 2 with low Bremer support, relationships among species in this clade differed somewhat among parsimonious trees. The monophyletic subclade comprising *P. elongatus* + (*P. iranicus + P. nisamiana*) is consistently recovered as sister to the remaining species in Clade 2. The subclade is supported as a monophyletic group by tiny lateral processes (char. 29: 2). Another subclade in Clade 2 is (*P. laevus* + *P. lusitanicus*) + ((*P. sahlbergii* + *P. angulaticeps*) + (*P. araxicus* + *P. modicus*)). This subclade is supported by the shark-fin-shaped caudal processes of the aedeagus (char. 31: 4), broad shaft in lateral view and short-spine-shapedlateral processes. The sister relationship between *P. laevus* and *P. lusitanicus* is supported by the partially fused base of the caudal processes of the aedeagus (char. 32: 2). The relationships within subclade (*P. sahlbergii* + *P. angulaticeps*) + (*P. araxicus* + *P. modicus*) are not consistently resolved among equally parsimonious trees.

Biogeographic analysis from the BBM method based on the MP tree in Figure 3 suggests that widespread Palearctic ancestors gave rise to lineages of *Planaphrodes* with more restricted distributions. The ancestor of Clade 1 was inferred to have a wide distribution across Eurasia but gave rise to a lineage of three species restricted to East Asia (*P. nigricans*, *P. baoxingensis* and *P. faciems*). In contrast, Clade 2 was inferred to have arisen in Europe and six species of this clade are still largely restricted to Europe. However, a lineage comprising two species (*P. elongatus* and *P. iranicus*) expanded its range into Western Asia, even Central Aisa.

### 3.2. Systematics

#### 3.2.1. Tribe Aphrodini

Notes. Aphrodini may be distinguished from other tribes of Aphrodinae by the following combination of traits: small- to medium-sized, robust, brown, somewhat depressed leafhoppers (Figures 4A–G, 5A–D and 8A–H), often with white bands or spots dorsally; head produced, with ocelli on anterior margin distant from ocelli and lateral frontal sutures extended ventromesad of ocelli; forewing veins elevated (Figure 11A); male with subgenital plates narrow at base, compressed distally (Figures 6E, 9D, 14C, 15C and 16C); style slender and sinuate (Figures 6F,G, 9E,F, 14D, 15D and 16D).

Aphrodini are similar to other Aphrodinae in the position of the ocelli and orientation of the lateral frontal sutures and structure of the male terminalia but differ in the relatively robust, depressed body form and elevated forewing veins (Figure 11A).

Hamilton [7] provided a key to genera of Aphrodini, as presently defined, which he treated as a subtribe of Aphrodini. Hamilton’s broad definition of Aphrodinae included most genera now included in Deltocephalinae [41], but Hamilton’s tribal classification was not adopted by subsequent authors. Dietrich [27] restricted the definition of Aphrodinae to include only Aphrodini, Portanini and Xestocephalini based on front femur row AV with single stout seta near midlength and female with ovipositor distinctly arcuate and later [42,43] added Sagmatiini and Euacanthellini to this subfamily as well. Further analyses are needed to confirm the monophyly of the subfamily and its included tribes.

#### 3.2.2. Genus *Planaphrodes* Hamilton, 1975

*Type species: Cicada tricincta* Curtis, 1836 [44].

Aphrodes (Planaphrodes), Anufriev, 1977 [14]; Anufriev et al., 1988 [9]: 168.

The original type species *Planaphrodes tricinctus* Curtis, 1836 [44] is confirmed a synonymy of *Planaphrodes bifasciatus* (Le Quesne, 1964) [8], which was placed in *Aphrodes* primordially.

Diagnosis. *Planaphrodes* differs from other genera of Aphrodini as follows: crown strongly depressed and produced, horizontal in profile (Figures 4A–G, 5A–D and 8A–H); aedeagal shaft compressed, lamellate; aedeagus with at least one pair of posteroventral processes arising in basal half of shaft (Figures 7A–H, 9G–J, 14E,F, 15E,F and 16E,F).

Description. (Modified from Hamilton 1975 [7,9]).

Small- to medium-sized leafhoppers, 3.0–6.0 mm in length. Coloration usually mostly brown to dark brown, with symmetrical white or yellow markings dorsally on head, pronotum and forewings sexually dimorphic and more extensive in males (Figures 4A–G, 5A–D and 8A–H).

Body robust, rugulose or shagreen. Head produced, longer medially than pronotum. Crown strongly flattened between ocelli, slightly to distinctly depressed laterally, anterior margins slightly upcurved, angularly rounded or parabolic in dorsal view and carinate to foliaceous in lateral view (Figures 4A–G, 5A–D and 8A–H). Coronal suture not reaching anterior margin, lateral frontal sutures extended to anterior margin of crown. Eyes broadly, shallowly notched next to antennal pit. Ocelli on anterior margin, visible dorsally, distant from eyes (Figures 4A–G, 5A–D and 8A–H). Face with frontoclypeus to anteclypeus flat to slightly inflated, frontoclypeus longer than wide with lateral margins expanded dorsad; lateral frontal sutures directed toward middle of ocelli; anteclypeus elongate with lateral margins slightly expanded medially and tapered apically (Figure 5I–O). Antennal ledges short, antennal pits moderately deep (Figure 5I–O). Pronotum with anterior margin roundly produced and posterior margin shallowly concave, sides short; surface shagreen. Scutellum and exposed part of mesonotum triangular, wider than long, slightly longer than pronotum (Figures 4A–G, 5A–D and 8A–H).

Forewings macropterous or submacropterous, with appendix very narrow; four apical cells present; outer subapical cell acute or truncate apically; inner subapical cell closed basally (Figure 11A,B). Fore femur with two stout anteroventral setae preapically, other setae small and irregularly arranged (Figure 12A–C). Setae on dorsal surface of fore and middle tibiae irregular. Hind femur with three apical macrosetae and one or two smaller preapical setae; hind tibial chaetotaxy PD 12, AD 9, AV 6, PV 6 (Figure 13A–F).

Pygofer usually sclerotized, taller than long, rounded caudal margin with distinct lobe or protrusion ventrally and hook-like process at middle of caudal margin directed medially (Figures 6A–D, 9A–C, 14A,B, 15A,B and 16A,B). Subgenital plates broad, curved, parallel-sided from base to rounded apex, with numerous irregularly arranged short setae (Figures 6E, 9D, 14C, 15C and 16C). Styles slender, with long crescent-shaped apical lobe denticulate on posteromesal margin. Connective short, Y-shaped, stem narrower and slightly longer than arms (Figures 6F–G, 9E,F, 14D, 15D and 16D). Aedeagal shaft strongly compressed, in lateral view narrowed near base, widened near middle, gradually narrowed distally, with two to four pairs of processes or spines in central section and a pair of retrorse processes near apex in most species (Figures 7A–H, 9G,H, 14F, 15F and 16F). Gonopore slit-shaped, on ventral surface above caudal spines (Figures 9I,J, 14E, 15E and 16E).

Female abdominal sternite VII broader than long, caudal margin concave medially (Figure 10C). Female pygofer with scattered setae over most of surface (Figure 10A,B). First valvulae, in lateral view, with dorsal margin nearly straight through most of length, somewhat narrowed near apex and tapered to apex, with dorsal preapical sculpture irregularly strigate, apical sculpture oblique, rugulose (Figure 10H,I). Second valvulae, in lateral view, with distal toothed blades occupying approximately two-fifths of total length, dorsal margin slightly elevated at apex of fused area, followed by slight concavity and second slight elevation at based of toothed distal part, teeth small and somewhat irregularly spaced over dorsal margin of evenly tapered distal section (Figure 10F,G). Third valvulae, in lateral view, with basal half narrow and apical half distinctly expanded, apex prominently rounded and extended slightly beyond pygofer, with sparse small setae ventrally (Figure 10D,E).

Distribution. Palaearctic region from Portugal to Japan.

Notes. We recognize the sixteen species, including two described as new, in the following checklist. Six species from China, Japan and Korea are described or re-described below based on comparative morphological study, and we propose three new synonymies. This study improves knowledge of the geographical distribution of the genus. As suggested by previous authors (e.g., Hamilton [7], Tishechkin [21]), the aedeagus is the most reliable character for distinguishing species. The key for the male genitalia of the *Planaphrodes* have been studied.

#### 3.2.3. Checklist of Species of *Planaphrodes* Hamilton

Distribution data, except for the new species, are summarized from Dmitriev et al. [1].

*Planaphrodes angulaticeps* (Emeljanov, 1964) [19].

Distribution. Romania, Russia, former Yugoslavia.

*Planaphrodes araxicus* (Logvinenko, 1971) [29].

Distribution. Azerbaijan.

*Planaphrodes baoxingensis* sp. nov.

Distribution. China (Sichuan).

*Planaphrodes bifasciatus* (Linnaeus, 1758) [45].

Distribution. Albania, Armenia, Austria, Azerbaijan, Belgium, Bulgaria, China, Czech, Denmark, Estonia, Finland, France, Germany, Greece, Hungary, Ireland, Israel, Italy, Kazakhstan, Kyrgyzstan, Latvia, Lithuania, Moldova, Netherlands, Norway, Poland, Portugal, Romania, Russia, Slovakia, Spain, Sweden, Switzerland, Ukraine, United Kingdom, Uzbekistan, former Yugoslavia.

*Planaphrodes elongatus* (Lethierry, 1876) [46].

Distribution. Armenia, Azerbaijan, Bulgaria, Czech, Georgia, Hungary, Italy, Kyrgyzstan, Romania, Russia, Slovakia, Syria, Ukraine, Uzbekistan, former Yugoslavia.

*Planaphrodes faciems* sp. nov.

Distribution. China (Hubei).

*Planaphrodes iranicus* (Dlabola, 1971) [31].

Distribution. Iran.

*Planaphrodes laevus* (Rey, 1891) [47].

Distribution. Austria, Belgium, Bohemia, Czech, Denmark, Estonia, Finland, France, Germany, Greece, Hungary, Italy, Kazakhstan, Kyrgyzstan, Latvia, Lithuania, Moldova, Mongolia, Moravia, Netherlands, Norway, Poland, Romania, Russia, Slovakia, Sweden, Switzerland, Ukraine, United Kingdom, Uzbekistan, former Yugoslavia.

*Planaphrodes lusitanicus* (Rodrigues, 1968) [30].

Distribution. Portugal.

*Planaphrodes modicus* (Logvinenko, 1966) [28].

Distribution. Moldova.

*Planaphrodes monticola* (Logvinenko, 1965) [48].

Distribution. Kazakhstan, Mongolia, Russia, Ukraine.

*Planaphrodes nigricans* (Matsumura, 1912) [11].

Distribution. China, Japan, Korea.

*Planaphrodes nigritus* (Kirschbaum, 1868) [49].

Distribution. Algeria, Austria, Azerbaijan, Belgium, Bulgaria, Czechoslovakia, France, Germany, Italy, Kazakhstan, Kyrgyzstan, Latvia, Moravia, Poland, Romania, Portugal, Sweden, Switzerland, Turkey, Ukraine, former Yugoslavia.

*Planaphrodes nisamiana* (Logvinenko, 1983) [17].

Distribution. Russia.

*Planaphrodes sahlbergii* (Signoret, 1879) [50].

Distribution. China, Japan, Korea, Mongolia, Russia.

*Planaphrodes vallicola* (Logvinenko, 1967) [51].

Distribution. Georgia, Russia.

#### 3.2.4. Key to Species of the Genus *Planaphrodes*


1.Aedeagal shaft with more than three pairs of processes (Figures 7A–F, 15F and 16F)...... 2-Aedeagal shaft with less than or equal to three pairs of processes (Figures 9G,H and 14F)...........................................................................................................................................62.Lateral processes of aedeagus situated higher than caudal processes (Figure 14F).... 3-Lateral processes of aedeagus situated lower than caudal processes (Figures 7A–F, 15F and 16F)...........................................................................................................................................43.Dorsal processes of aedeagus situated higher than lateral processes..............*P. nigritus*-Dorsal processes of aedeagus situated lower than lateral processes........... *P. monticola*4.Aedeagal shaft with five pairs of retrorse processes (Figure 16F).....*P. faciems* sp. nov.-Aedeagal shaft with four pairs of retrorse spines (Figures 7A–F and 15F)..................55.Dorsal processes of aedeagus long and strongly divergent, curved laterally (Figure 7G,H).................................................................................................................... *P. nigricans*-Dorsal processes of aedeagus short, not strongly divergent, directed posteriorly (Figure 15E).............................................................................................. *P. baoxingensis* sp. nov.6.Aedeagal shaft with two pairs of retrorse processes......................................... *P. vallicola*-Aedeagal shaft with three pairs of retrorse processes (Figures 9G,H and 14F)............. 77.Aedeagal shaft with apical denticles.............................................................. *P. bifasciatus*-Aedeagal shaft with apical spines (Figures 7A–F, 9G,H, 14F, 15F and 16F).................. 88.Aedeagal shaft slender in lateral view, slightly widened near middle......................... 9-The width of aedeagal shaft moderate in lateral view.................................................... 119.Lateral processes of aedeagus situated higher than caudal processes.......... *P. elongatus*-Lateral processes of aedeagus situated lower than caudal processes........................... 1010.Apical retrorse spines of aedeagus tiny, hook-like........................................... *P. iranicus*-Apical retrorse spines of aedeagus strongly widened, hook-like................. *P. nisamiana*11.Aedeagal shaft with almost uniform width in lateral view (Figure 9G,H).................12-Aedeagal shaft apical 1/3 obviously tarpered in lateral view (Figure 14F).................1312.Aedeagus with apical retrorse spines arising ventrally (Figure 9G,H)....... *P. sahlbergii*-Aedeagus with apical retrorse spines arising laterally.................................... *P. araxicus*13.Aedeagus with caudal process short and poorly developed........................... *P. modicus*-Aedeagus with caudal process shark fin-like and well developed............................... 1414.Aedeagus with apical retrorse spines strongly widened hook-like......... *P. angulaticeps*-Aedeagus with apical retrorse spines tiny hook- like.................................................... 1515.Aedeagus with apical spines directed ventrad down................................... *P. lusitanicus*-Aedeagus with apical spines directed dorsad down (Figure 14F)..................... *P. laevus*


### 3.3. Taxonomy

#### 3.3.1. *Planaphrodes nigricans* (Matsumura, 1912) (Figure 4A–I, Figure 5M, Figure 6A–G and Figure 7A–H)

*Acocephalus bifasciatus* var. *nigricans* Matsumura, 1912: 289 [11].

*Acocephalus nigricans* Kato, 1933b: 27 [52].

*Aphrodes nigricans* Ishihara, 1953: 35 [53]; Esaki and Ito, 1954: 83 [54]; Metcalf, 1963: 185 [55]; Nast, 1972: 239 [56].

*Aphrodes bifasciatus nigricans* Ishihara, 1965: 125 [57].

*Planaphrodes sahlbergi* (nec Signoret) Hamilton, 1975: 1012 [7].

*Planaphrodes sahlbergi* (nec Signoret) Lee and Kwon, 1977: 66 [58].

*Aphrodes (Planaphrodes) nigricans* Anufriev, 1978: 56 [15]; Anufriev et al., 1988: 165 [9].

*Planaphrodes nigricans* Lee and Kwon, 1979: 154 [59]; Hayashi et al., 2016: 282 [25].

*Planaphrodes bella* Choe, 1981: 152 [16]. syn. nov.

Description. Length (including wings): male 5.5–5.7 mm, female 5.7–5.9 mm.

Male: coloration and markings of body highly variable. Crown black, with several unevenly smallyellow spots close to anterior margin variably developed. Ocelli white (Figure 4A–G). Face mostly pale yellow, anteclypeus slightly darker. Eyes greyish dark blue (Figure 5M). Pronotum with anterior half black, whitish to pale brown in posterior half. Scutellum dark brown with apex somewhat paler. Forewings normal or submacropterous, brown, opaque, veins concolorous with cells, some specimens with three large irregular whitish transverse bands, which may be reduced to spots situated in basal, median and subapical region, respectively (Figure 4A–G). Abdomen brown. Legs brown, with brown macrosetae (Figure 4H). Female: greyish brown, finely dark mottled, with more or less distinct dark spots along costal margin of forewings. Face generally paler than upper side of body (Figure 4I).

Pygofer well sclerotized, taller than long, caudal margin slightly concave, with distinct round posteroventral lobe covered with tiny setae and hook-like process arising at middle of caudal submargin directed medially with several denticles (Figure 6A–D). Subgenital plates broad, curved, parallel-sided in basal half in ventral view, narrowed to rounded apex, with numerous irregularly arranged short setae (Figure 6E). Styles slender, with long, slightly broadened crescent-shaped apical lobe irregularly dentate on ventromesal margin. Connective distinctly longer than wide (Figure 6F,G). Aedeagal shaft strongly compressed, in lateral view narrowed in its basal portion, widened near middle, gradually narrowing in its distal part, with four pairs of retrorse processes, apical spines tiny and hook-like at apex; dorsal processes long and strongly divergent, curved laterally; caudal processes directed caudally; lateral processes small, slender and directed ventrally (Figure 7A–F). Gonopore slit-shaped on posterior surface above caudal spines (Figure 7G,H).

Material examined. Holotype of *Planaphrodes bella* Choe; KOREA·1♂; JN, Soheuksando; 11 August 1974; K.R. Choe leg.; KNU. KOREA—**Jeollanam-Do** 1♂; Jindo; 17 July 1984; Y.J. Kwon leg.; KNU·1♀; Mudeungsan, 26 July 1981; Y.J. Kwon leg.; KNU·1♂; Wando; 17 June 2003; Y.J. Kwon leg.; KNU—**Jeju-Do**·4♂, 11♀; Hallasan; 6–11 August 1984; Y.J. Kwon leg.; KNU·3♂, 3♀; Hallasan; 5 August 1989; Y.J. Kwon leg.; KNU·1♂; Hallasan; 7 September 1998; Y.J. Kwon leg.; KNU—**Chungcheongnam-Do**·1♂; Deoksungsan; 27 May 1982; Y.J. Kwon leg.; KNU—**Gyeongsangbuk-Do**·1♂; Palgongsan; 24 August 1980; Y.J. Kwon leg.; KNU. **Lectotype** designated by K.G.A. Hamilton; JAPAN·1♂; “LECTOTYPE, *Acocephalus nigricans* Matsumura”; Moji; 29 June 1903; “Det. KGAH ‘74”; SEHU. **Paralectotype** 1♂, same data as for lectotype; JAPAN—**Honshu**·3♂ 2♀; Akita Pref., Yuzawa, Takamatsu; 19 August 2010; M. Hayashi et al. leg.; ELKU·4♂2♀; Yamagata Pref., Asahi, Tachiki; 20 August 2010; M. Hayashi et al. leg.; ELKU·2♂ 2♀; Fukushima Pref., Aizu-wakamatsu, Higashiyama; 21 August 2010; M. Hayashi et al. leg.; ELKU·2♂; Gumma Pref., Kawayu, Mae-hotaka; 13 July 2008; K. Yoshida leg.; ELKU·4♂ 2♀; Saitama Pref., Mt. Jomine; 28 July 1983; M. Hayashi et al. leg.; ELKU·1♂ 20♀; Saitama Pref., Oku-Chichibu Mts., Mikuni Pass/Akuseki (1760–1850 m); 15 September 1982; M. Hayashi et al. leg.; ELKU·20♂ 14♀; Saitama Pref., Ranzan, Shogunzawa; 24 June 2010; M. Hayashi et al. leg.; ELKU·1♂; Toyama City, Awasuno; 21 July 1954; S. Takagi leg.; ELKU·3♂ 3♀; Nagano Pref., Sugadaira (1330 m); 2 August. 1996; M. Hayashi et al. leg.; ELKU·1♂; Okayama Pref., Hayama Valley; 18 June 2001; T. Nozaki leg.; ELKU—**Shikoku**·1♂ 1♀; Ehime Pref., Uchiko, Nakagawa; 7 July 2004; M. Hayashi et al. leg.; ELKU—**Kyushu**·1♂; Mt. Fukuoka Pref., Sefuri; 27 August 2004; M. Hayashi et al. leg.; ELKU·3♂; Kagoshima Pref., Minami-osumi, Sata; 29 May 1952; H. Hasegawa leg.; NIAES·2♂ 1♀; Nagasaki pref., Tsushima Is., Izuhara, Kamizaka; 12 July 1995; M. Hayashi et al. leg.; ELKU·1♂; Nagasaki pref., Tsushima Is., Mitsushima, Kusugahama; 13 July 1995; M. Hayashi et al. leg.; ELKU.CHINA·2♂; Heilongjiang Prov., Pine forest in Sanjiang plain wetland; 17 August 2007; X. Bao and L. Wei leg.; NWAFU.

Distribution. China (Heilongjiang), Japan (Hokaido, Honshu, Shikoku, Kyushu, Tsushima Island), Korea (Jejudo), Mongolia, Russia (Maritime Territory).

Diagnosis. This species was originally described by Matsumura [11] from Japan as a variety of *Acocephalus bifasciatus*. It was raised to species level as *Acocephalus nigricans* by Kato [52] and transferred to *Aphrodes* by Ishihara [53]. Hamilton [7] checked the type specimens of *P. nigricans* and synonymized it with *P. sahlbergi* (Signoret, 1879) [50]. Nevertheless, Emeljanov [10] indicated that *P. sahlbergi* as treated by Hamilton [7] was an error for *P. nigricans*. Meanwhile, Anufriev [15] determined that *P. nigricans* and *P. guttatus* were confused by Hamilton [7] and *P. nigricans* was indeed valid. One of us (Hayashi) checked and investigated the type series in the Matsumura Collection (SEHU), including the lectotypes designated by Hamilton, and confirmed that *P. nigricans* and *P. guttatus* were confused by Hamilton [7] and that *P. nigricans* is valid. We also photographed the syntypes of *P. bella* in Figure 7A–E; the identical genitalia illustrated that *P. bella* is a junior synonym of *P. nigricans*. Here, this species is recorded in China for the first time.

We found the aedeagus in specimens from China, Korea and Japan not only distinctly different from those illustrated by Anufriev et al. [9] but also quite variable among each other. These differences, which we interpret as intraspecific variation, occur mainly in the shape and proportions of the aedeagus shaft.

**Figure 4 insects-14-00291-f004:**
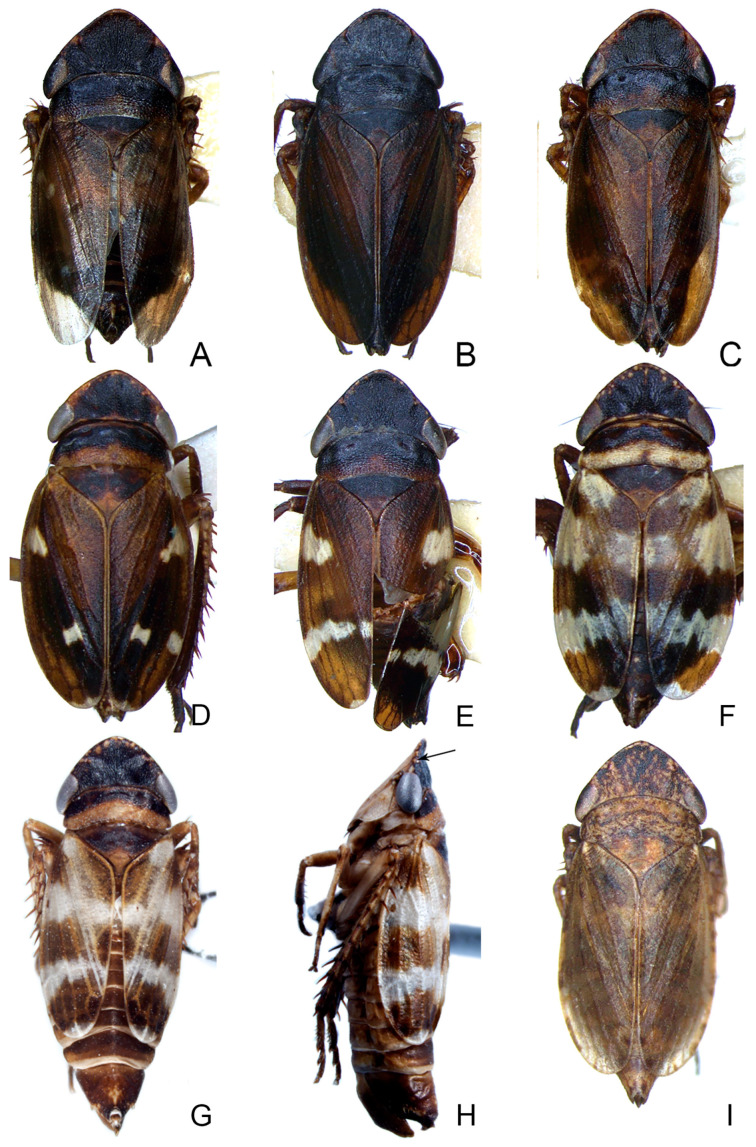
(**A**–**I**) *Planaphrodes nigricans*, dorsal and lateral habitus. (**A**) ♂, Korea, Deogsungsan CN. (**B**) ♂, Korea, Hallasan JJ. (**C**) ♂, Korea, Wando JN. (**D**) ♂, Korea, Hallasan JJ. (**E**) ♂, Korea, Jindo JN. (**F**) ♂, Korea, Hallasan JJ. (**G**,**I**) ♂, China, Heilongjiang. (**H**) ♀, Korea, Hallasan JJ.

**Figure 5 insects-14-00291-f005:**
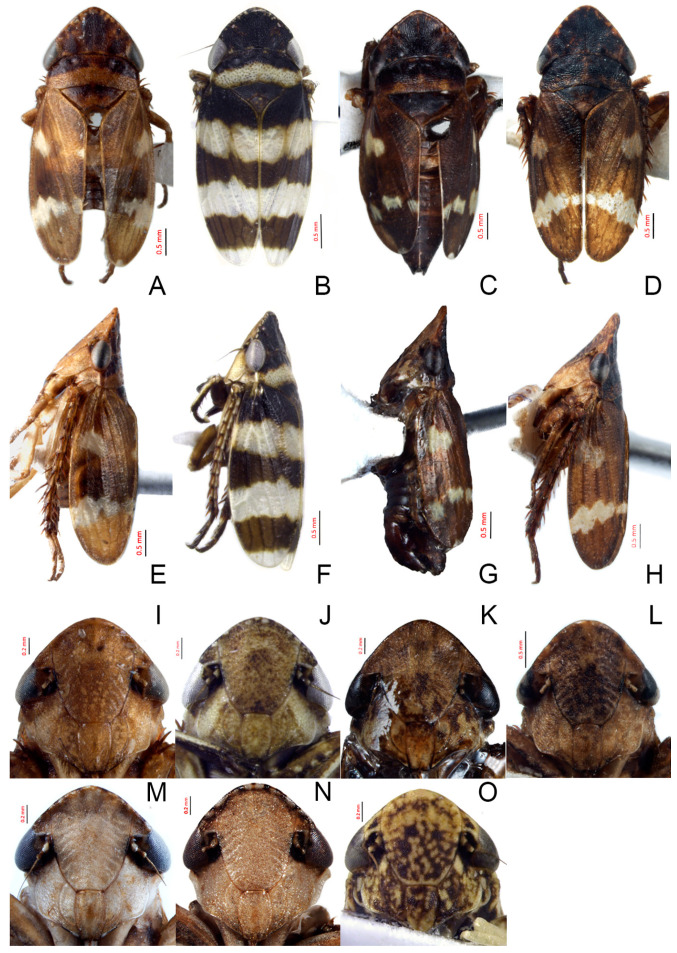
(**A**–**O**) *Planaphrodes* spp., male dorsal, lateral habitus and face. (**A**,**E**,**I**) *P. bifasciatus*. (**B**,**F**,**J**) *P. laevus*. (**C**,**G**,**K**) *P. arundosis* sp. nov. (**D**,**H**,**L**) *P. faciems* sp. nov. (**M**) *P. nigricans*. (**N**) *P. sahlbergii* (China, Jilin). (**O**) *P. sahlbergii* (Qinghai, China).

**Figure 6 insects-14-00291-f006:**
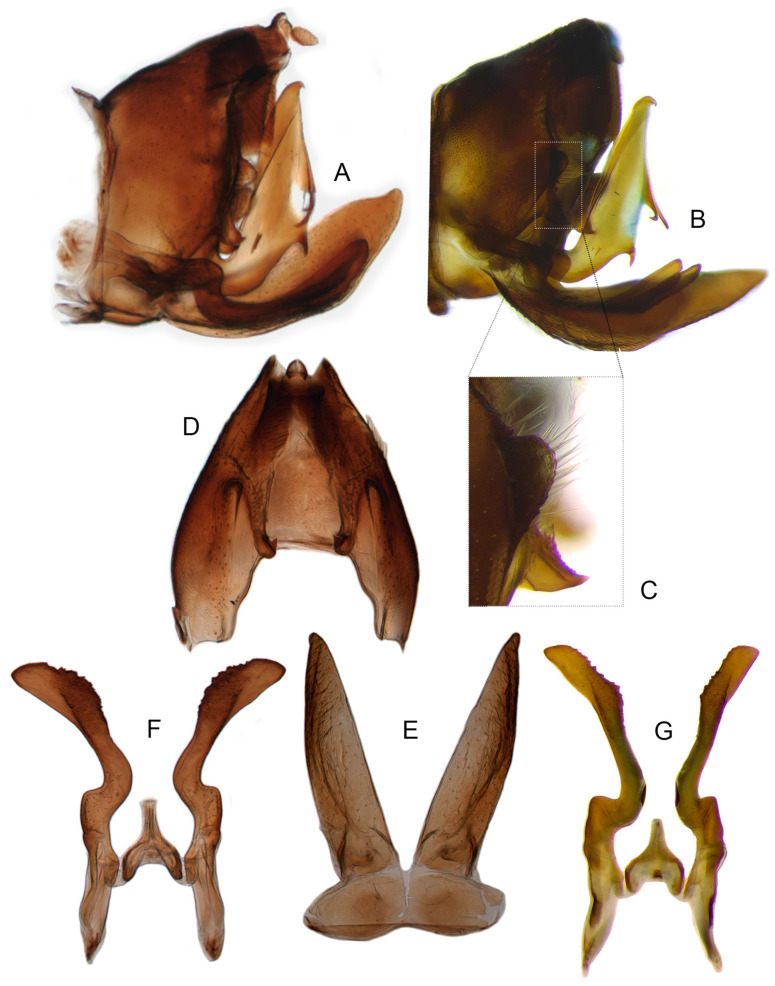
(**A**–**G**) *Planaphrodes nigricans*. (**A**,**B**) pygofer, lateral view. (**C**) caudal margin processes of pygofer. (**D**) genital capsule, ventral view. (**E**) subgenital plates, ventral view. (**F**,**G**) styles and connective, ventral view (**A**,**D**–**F**) from China, Heilongjiang Province; (**B**,**C**,**G** from Korea, Jindo JN).

**Figure 7 insects-14-00291-f007:**
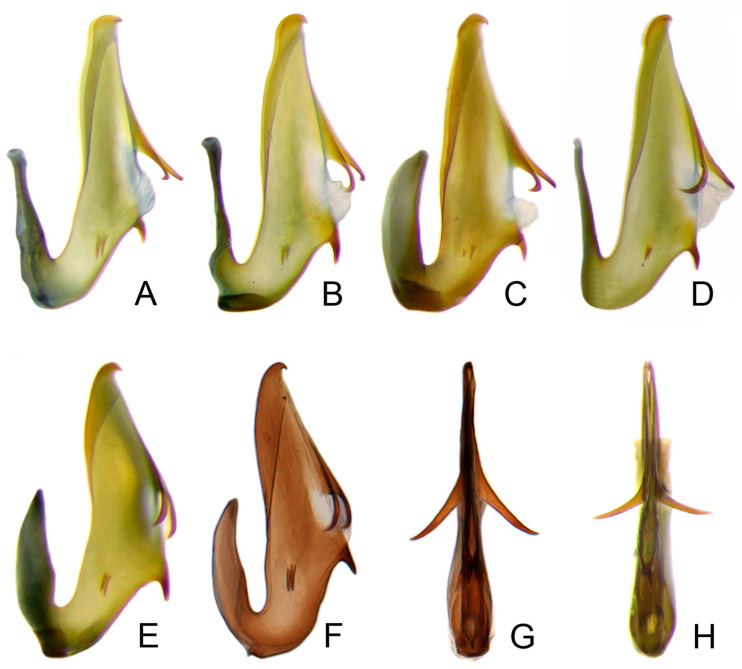
(**A**–**H**) *Planaphrodes nigricans*, aedeagus in lateral and caudal view. (**A**) Korea, Hallasan JJ. (**B**) Korea, Deogsungsan CN. (**C**) Korea, Jindo JN. (**D**) Korea, Wando JN. (**E**) Korea, Palgongsan GB. (**F**) China, Heilongjiang. (**G**) China, Heilongjiang. (**H**) Korea, Jindo JN.

#### 3.3.2. *Planaphrodes Sahlbergii* (Signoret, 1879) (Figure 5N,O, Figure 8A–K, Figure 9A–J and Figure 10A–I)

*Acocephalus sahlbergii* Signoret, 1879: 72 [50].

*Acocephalus sahlbergi* Puton, 1886: 80 [60]; Oshanin, 1906: 91 [22]; Oshanin, 1912: 103 [61].

*Acocephalus bifasciatus* var. *guttatus* Matsumura, 1912: 289 [11].

*Acocephalus alboguttatus* Kato, 1933c: 8 [23] syn. nov.

*Acocephalus guttatus* Kato, 1933b: 27 [52].

*Aphrodes sahlbergi*, Jacobi, 1943: 31 [24]; Metcalf, 1963: 187 [55].

*Aphrodes japonicus* Dlabola, 1960: 240 [12].

*Aphrodes mongolicus* Dlabola, 1965: 100 [13].

*Aphrodes guttatus*, Nast, 1972: 238 [56]; Hayashi et al., 2016: 282 [25]; Okada, 1976 [62]: 187; Nast, 1972: 238 [56].

*Planaphrodes sahlbergi*, Hamilton, 1975 [7]: 1014; Anufriev, 1978: 56 [15]; Lee, 1979: 348 [63]; Lee and Kwon, 1979: 858 [59]; Morimoto, 1989: 98 [64]; Kim et al., 1994: 86 [65]; Kwon and Huh, 2001: 154 [26].

*Planaphrodes mongolica*, Lee and Kwon, 1977: 66 [58].

*Aphrodes* (*Planaphrodes*) *sahlbergi*, Anufriev et al., 1988 [9]: 165.

*Aphrodes daiwenicus* Kuoh, 1981 [66]; Kuoh, 1992 [67]; Cai and Shen, 2002 [68]: 274 syn. nov.

Description. Length (including wings): male 5.0–5.4 mm.

Male: crown dark brown, with several small yellow spots near anterior margin (Figure 8A–H). Face brown, anteclypeus with rather indistinct dark mottling. Eyes grey. Pronotum dark brown with broad ivory white transverse band in posterior margin. Scutellum dark brown (Figure 5N,O). Forewings pale brown, opaque, veins dark brown, with a large irregular whitish or yellowish transverse band and a whitish or yellowish cloudy patch situated in subapical and basal 1/3 region, respectively, apex yellowish; some whitish or yellowish coloration quite extended (Figure 8A–H). Abdomen pale brown. Legs pale brown, with brown macrosetae (Figure 8K). Female: yellowish brown, above and below more or less densely dark mottled (Figure 8I,J).

Pygofer well sclerotized, taller than long, caudal margin slightly concave, with distinct posteroventral lobe covered with tiny setae and hook-like process arising at middle of caudal submargin directed medially with apical denticles (Figure 9A–C). Subgenital plates broad, curved, parallel-sided at basal half in ventral view, narrowed to rounded apex, with numerous irregularly arranged short setae (Figure 9D). Styles slender, with long paddle-shaped apical lobe evenly broadened and denticulate on ventromesal margin. Connective nearly as wide as long (Figure 9E,F). Aedeagal shaft straight, with caudal margin slightly concave, bearing three pairs of retrorse processes, apical processes tiny, directed caudally and then curved ventrally; ventral spines tapering apically and directed caudoventrally, moderately long and slightly divergent; lateral processes short, directed ventrally (Figure 9G,H). Gonopore slit-shaped, on ventral surface at basal 1/3 above caudal spines (Figure 9I,J).

Female abdominal sternite VII nearly twice as wide as long, caudal margin broadly concave, with narrow V-shaped medial indentation (Figure 10C).

Material examined. Holotype of *Acocephalus alboguttatus* Kato; CHINA·1♂; Manchuria, Andoken; 26 June 1932; K. Kikuchi leg. UMUT. CHINA·1♂; Sichuan Prov., Ruoergai prairie, Dazhasi; 3380 m; 23 July 1963; L.Y. Zheng leg.; NKU·1♂; Gansu Prov., Wenxian, Yanggashan; 2000 m; 3 July 2001; Q. Sun leg.; NWAFU·2♂; Heilongjiang Prov., Pine forest in Sanjiang plain wetland; 17 August 2007; X. Bao and L. Wei leg.; NWAFU·1♂; Hebei Prov., Weicheng, Hongsongwa; 11 August 2006; Y.N. Duan leg.; NWAFU·1♂; Jilin Prov., Erdaobaihe peace forest station; 738 m; 22 July 2006; X.M. Zhang leg.; NWAFU. JAPAN—**Hokkaido**·2♂; Shiretoko Pen., 2 August 1959; H. Fukushima et al. leg.; NIAES·2♀; Akanuma, Kushiro Marsh, Hokkaido, 28, VIII, 1990, M. Hayashi et al. leg.; ELKU·1♂; Horokanai; 30 July 1958; K. Kamijo leg.; SEHU·1♂; Sapporo, Kotoni; 2 July 1959; H. Hasegawa leg.; NIAES·1♂; Sapporo, Jozankei/Asari Pass; 13 July 1985; M. Hayashi leg.; ELKU—**Honshu**·1♂; Saitama Pref., Sakado, Higashi-wada; 12 June 2001; M. Hayashi et al. leg.; ELKU·1♂; Nagano Pref., Sugadaira (1330 m); 8 July 2004; M. Hayashi et al. leg.; ELKU·1♂.

**Figure 8 insects-14-00291-f008:**
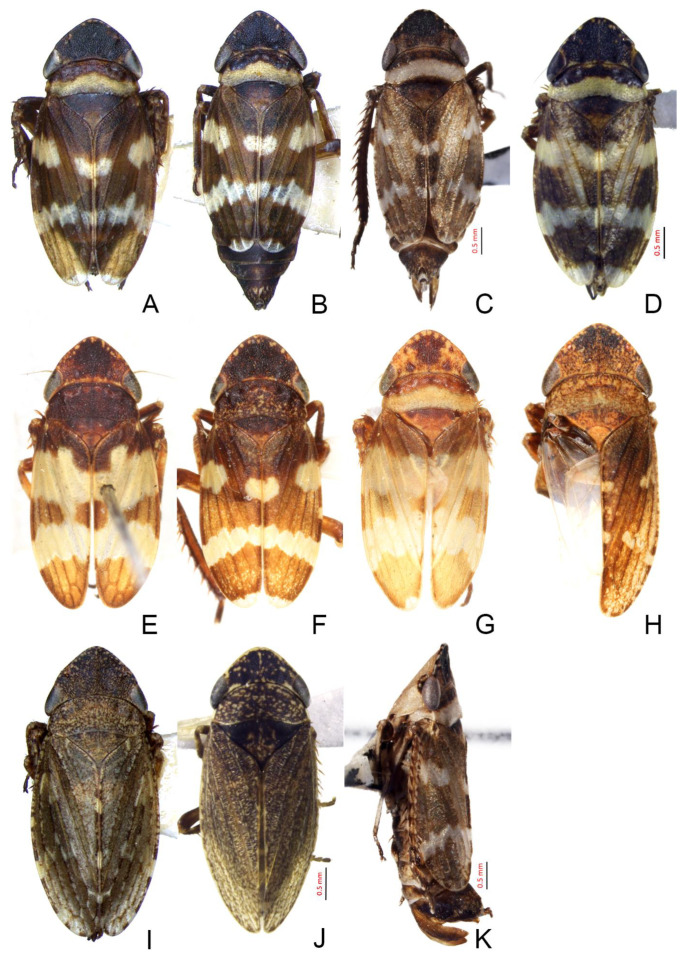
(**A**–**K**) *Planaphrodes sahlbergi*, dorsal and lateral habitus. (**A**) ♂, Korea, Seolaksan GW. (**B**) ♂, Korea, Odaesan GW. (**C**) ♂ China, Jilin. (**D**) ♂ China, Qinghai. (**E**) ♂, Japan, Hokkaido, Horokanai. (**F**) ♂, Japan, Honshu, Sakado. (**G**) ♂, Japan, Hokkaido Shiretoko Pen. (**H**) ♂, Japan, Honshu, Okayama. (**I**) ♀ Korea, Seolaksan GW. (**J**) ♀ China, Qinghai. (**K**) ♂ China, Jilin.

**Figure 9 insects-14-00291-f009:**
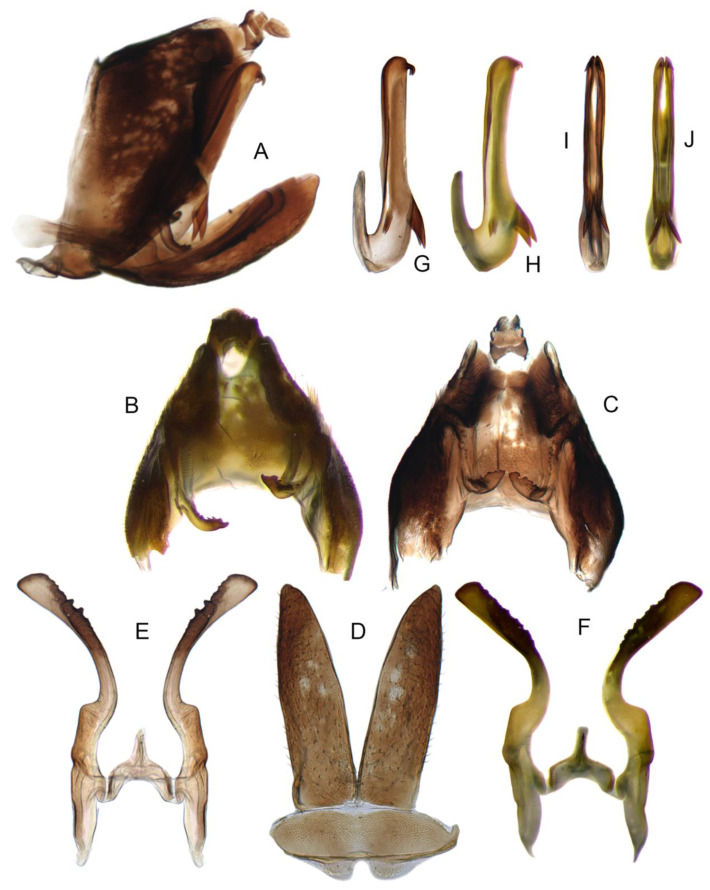
(**A**–**J**) *Planaphrodes sahlbergii*. (**A**) pygofer, lateral view. (**B**,**C**) genital capsule, ventral view. (**D**) subgenital plates, ventral view. (**E**,**F**) stylets and connective, ventral view. (**G**,**H**) aedeagus, lateral view. (**I**,**J**) aedeagus, ventral view (**A**–**E**,**G**,**I**) from China, Jilin; (**B**,**F**,**H**,**J**) from Korea, Seolaksan GW).

**Figure 10 insects-14-00291-f010:**
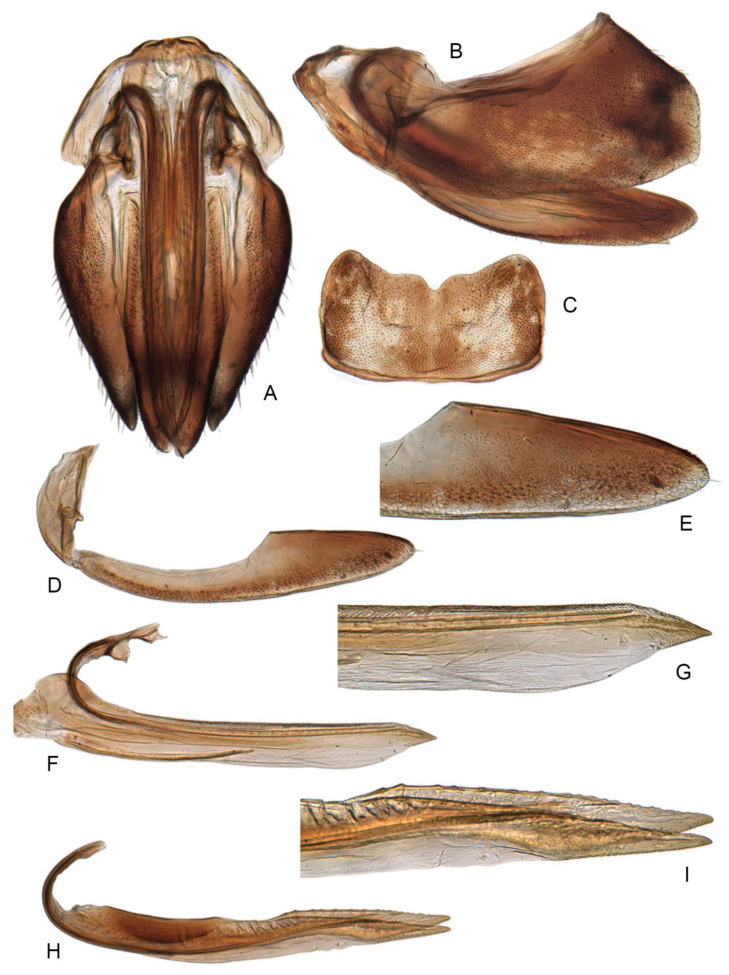
(**A**–**I**) Female genitalia of *Planaphrodes sahlbergii*. (**A**) ventral view. (**B**) lateral view. (**C**) abdominal sternum VII, ventral view. (**D**) third valvulae. (**E**) apex of third valvulae. (**F**) first valvulae. (**G**) apex of first valvulae. (**H**) second valvulae. (**I**) apex of second valvulae.

Okayama, Chikko-Midorimachi; 28 August 2009; K. Ikeda leg.; ELKU.KOREA—**Gyeongsangbuk-Do**·1♂; Byeongpungsan; 13 August 1997; Y.J. Kwon leg.; KUN·2♂ 3♀; Juwangsan; 19 Jul 1981; Y.J. Kwon leg.; KUN·2♂; Juwangsan; 26 July 1984; Y.J. Kwon leg.; KUN·2♀; Seonuisan; 16 July 1997; Y.J. Kwon leg.; KUN—**Gangwon-Do**·1♂; Odaesan; 4 August. 1983; Y.J. Kwon leg.; KUN·1♀; Seoraksan; 2 July 1984; Y.J. Kwon leg.; KUN·3♂ 3♀; Seoraksan; 27–28 Jul. 1982; Y.J. Kwon leg.; KUN—**Gyeongsangnam-Do**·1♂; Geumjeongsan; 16 October. 1983; Y.J. Kwon leg.; KUN—**Gyeonggi-Do**·1♂; Sineup; 20 July 2006; Y.J. Kwon leg.; KUN.

Distribution. China (Gansu, Hebei, Heilongjiang, Jilin, Xizang), Japan (Hokkaido, Honshu, Shikoku), Korea (Ulleungdo), Mongolia, Russia (E Siberia, Primorskiy kray).

Diagnosis. This species was originally described by Signoret [50] from Daoli (Daourie), China. Although the status was constantly changed [7,15], *P. sahlbergii* was valid. Recently, one of us (Hayashi) investigated the holotype (UMUT) of *Acocephalus alboguttatus* Kato, 1933 and found it is really a male, not a female as indicated in Kato’s original description. The specimen is, however, not in good condition, being a teneral one formerly infected by fungi and with the abdominal terminalia partly broken. However, Kato’s species is surely synonymous with *P. sahlbergii* based on the habitus, wing markings, coloration of legs and configuration of male genitalia. Kuoh [66] described *Aphrodes daiwenicus* from Xizang, China based on one male specimen. Comparing the original descriptions and illustrations with the syntypes of *A. daiwenicus* and *P. sahlbergii*, it is clear that these species are synonyms.

Previous authors disagreed over the correct spelling of the species name. Signoret [50] named the species in honor of another entomologist, J. Sahlberg using the ending -*ii*. Some subsequent authors used the genitive ending -*i* [7,9,22,24,68]. According to ICZN Article 33.4, the latter spelling is deemed to be an incorrect subsequent spelling and only the original species name *sahlbergii* published by Signoret [50] is valid (and available).

#### 3.3.3. *Planaphrodes bifasciatus* (Linnaeus, 1758) (Figure 5A,E,I, Figure 11A,B, Figure 12A–C and Figure 13A–F)

*Cicada bifasciata* Linnaeus, 1758: 436 [45].

*Cicada trifasciata* De Geer, 1773: 186 [69].

*Cercopis bifasciata*, Fabricius, 1775: 689 [70].

*Cicada tristriata* Gmelin, 1789: 2216 [71].

*Aphrophora bifasciata var. a Germar*, 1821: 51 [72].

*Jassus obliquus* Germar, 1821: 89 [72].

*Tettigonia bifasciata*, Curtis, 1829: 193 [73].

*Aphrodes bifasciata*, Curtis, 1829: 193 [73]; Li and Wang, 1991: 146 [74].

*Acucephalus bifasciatus*, Herrich-Schäffer, 1834 [75]: 1; Wu, 1935: 79 [76].

*Acucephalus albifrons* Herrich-Schäffer, 1835: 72 [77].

*Acucephalus tricinctus* Curtis, 1836: 620 [44].

*Pholetaera bifasciata*, Zetterstedt, 1840: 289 [78].

*Ptyelus bifasciatus*, Amyot and Serville, 1843: 567 [79].

*Acocephalus bifasciatus*, Marshall, 1865: 146 [80].

*Acocephalus granulatus* Fieber, 1872: 10 [81].

*Acocephalus albifrons bifasciatus Signoret*, 1879: 79 [50].

*Acocephalus befasciatus* Bachmetjew, 1901: 251 [82].

*Acocephalus trincitus* Bierman, 1907: 115 [83].

*Acocephalus tritinctus* Hofmänner, 1924: 53 [84].

*Aphrodes bifasciatus*, Blöte, 1927: 55 [85]; Sun, 1988: 19 [86].

*Philaenus bifasciatus*, Lallemand, 1949: 7 [87].

*Acocephalus obliqus* Metcalf, 1963: 253 [55].

*Planaphrodes tricinctus*, Hamilton, 1975 [7]: 1012; Tishechkin, 2019: 231 [21].

*Planaphrodes bifasciata*, Hamilton, 1975 [7]: 1012; Ossiannilsson, 1981 [3]: 367; Nast, 1987: 582 [88]; Zhang, 1990: 70 [89].

Description. Length (including wings): male 4.8 mm.

Crown dark brown, with some small yellow spots near anterior margin (Figure 5A). Face brown, anteclypeus with rather indistinct dark mottling. Eyes grey. Pronotum dark brown with broad ivory white transverse band on posterior margin(Figure 5I). Scutellum dark brown with a shallow spot at apex (Figure 5A). Forewings pale brown, opaque, veins dark brown, with large irregular yellowish transverse band and yellowish cloudy patch situated in subapical and basal 1/3, respectively, apex yellowish (Figure 11A). Abdomen pale brown. Legs pale brown, with brown macrosetae (Figure 5E). Female yellowish brown, above and below more or less densely dark mottled.

**Figure 11 insects-14-00291-f011:**
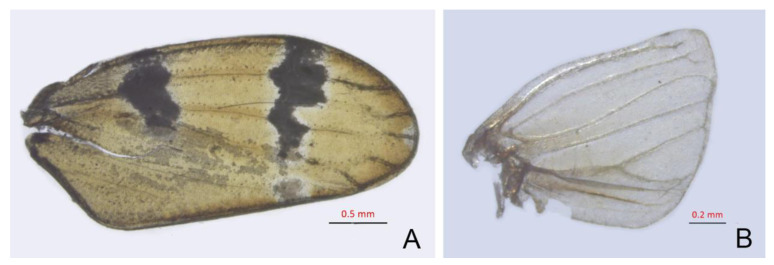
(**A**,**B**) *Planaphrodes bifasciatus* ♂. (**A**) forewing. (**B**) hind wing.

**Figure 12 insects-14-00291-f012:**
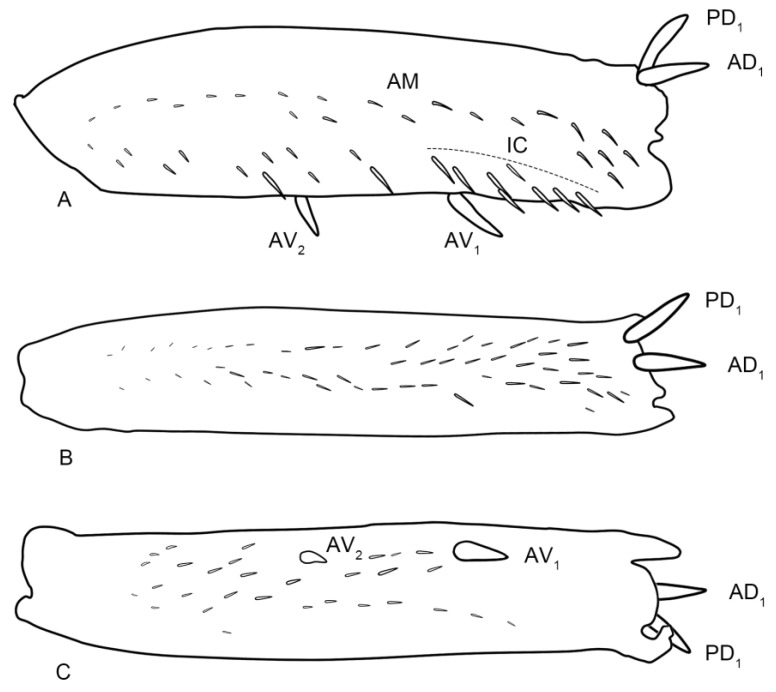
(**A**–**C**) The left forefemora of *Planaphrodes bifasciatus*. (**A**) anterior view. (**B**) dorsal view. (**C**) ventral view. AM = anteromedial row; IC = intercalary row; AV = anteroventral row; AD = anterodorsal row; PD = posterodorsal row.

**Figure 13 insects-14-00291-f013:**
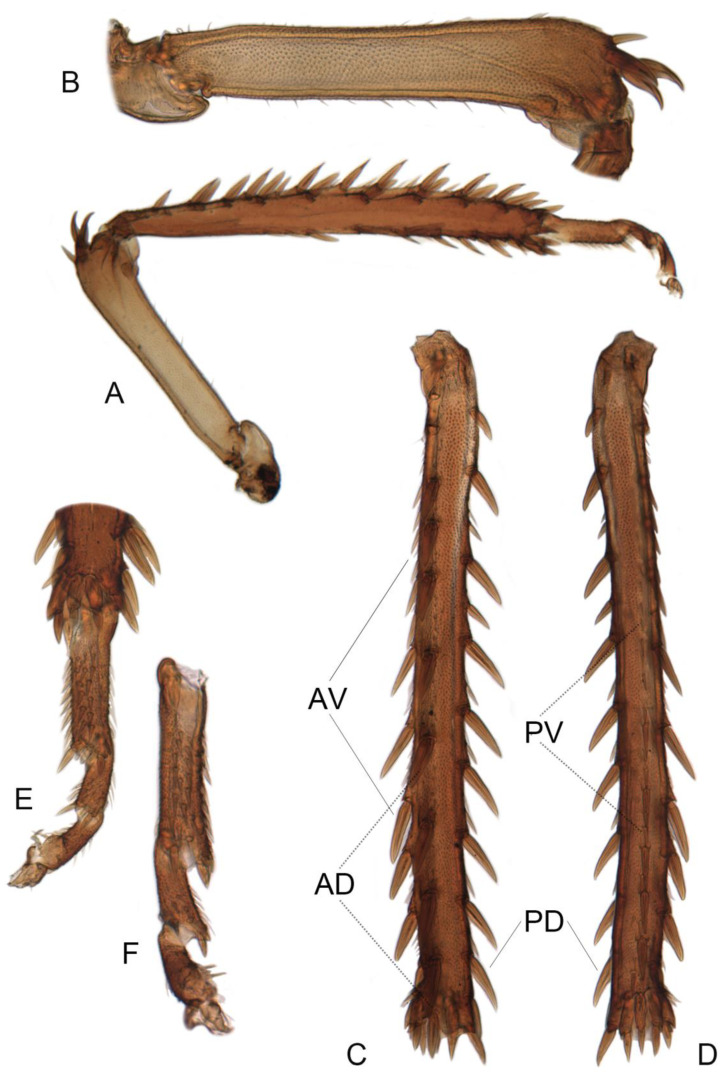
(**A**–**F**) *Planaphrodes bifasciatus* ♂. (**A**) right hind leg, anterior view. (**B**) right hind femur, anterior view. (**C**) right hind tibia, dorsal view. (**D**) right hind tibia, ventral view. (**E**) right hind tarsus, anterior view. (**F**) right hind tarsus, posterior view. AD = anterodorsal row; AV = anteroventral row; PD = posterodorsal row; PV = posteroventral row.

Aedeagal shaft broad in lateral view, with three pairs of retrorse processes of same length, including two pairs of processes near caudal margin in central section (a pair of dorsal processes and a pair of caudal processes) and a pair of processes on sides closer to anterior margin (lateral processes); apex simple, rounded.

Material examined. CHINA·1♂; Jilin Prov., Linjiang, Naozhi; 24 July 1983; B.Z. Hua and Z.L. Wu leg.; NWAFU.

Distribution. China (Gansu, Guizhou, Jilin), Albania, Armenia, Austria, Azerbaijan, Belgium, Bohemia, Bulgaria, Czech, Denmark, Estonia, Finland, France, Germany, Greece, Hungary, Ireland, Israel, Italy, Kazakhstan, Kyrgyzstan, Latvia, Lithuania, Moldova, Moravia, Netherlands, Norway, Poland, Portugal, Romania, Russia, Slovakia, Spain, Sweden, Switzerland, Ukraine, United Kingdom, Uzbekistan, former Yugoslavia.

Diagnosis. This species is widely distributed in the Palearctic Region. Oshanin [22] first recorded this species from China. Recently, Zhang [89] recorded this species from China based on one male specimen and provided the illustrations of external habitus and aedeagus. Unfortunately, we cannot provide photos of the male genitalia but refer to the illustration of the male genitalia from Ribaut [36] as the dissection of the only male Chinese specimen is missing the male genitalia. It closely resembles the Palaearctic species *Planaphrodes nigritus* (Kirschbaum) from which it differs in the shape of the aedeagus and apical process of the shaft.

#### 3.3.4. *Planaphrodes laevus* (Rey, 1891) n. rec from China (Figure 5B,F,J and Figure 14A–F)

*Aphrodes trifasciata* Curtis, 1829:193 [73].

*Acucephalus trifasciatus*, Curtis, 1836:620 [44].

*Acocephalus trifasciatus*, Eversmann, 1837:33 [90].

*Athysanus trifasciatus*, Yersin, 1856:748 [91].

*Acocephalus trifasciatus var. laevus* Rey, 1891:245 [47].

*Acocephalus trifasciata var. niger* Kusnezov, 1929:176 [92].

*Aphrodes turkestanicus* Dubovsky, 1966:84 [93].

*Aphrodes griseus* Mitjaev, 1967:719 [94].

*Planaphrodes laevus*, Hamilton, 1975 [7]; Tishechkin, 2019: 228 [21].

*Aphrodes laevus* Koçak, 1981:41 [95].

Description. Length (including wings): male 4.7 mm.

Crown black, flattened and rugose. Ocelli white (Figure 5B). Face light brown. Eyes grey (Figure 5J). Pronotum black, with white band on posterior margin. Scutellum dark. Forewings black, opaque, veins distinct, three white bands intersect with three black bands (Figure 5B). Legs dark brown (Figure 5F).

Pygofer well sclerotized, taller than long, caudal margin slightly concave, with distinct round lobe covered with tiny setae and a hook-like process situated at middle of caudal submargin directed mesad (Figure 14A,B). Subgenital plates broad, curved, parallel-sided at basal half in ventral view, narrowed to rounded apex, with numerous irregularly arranged short setae (Figure 14C). Styles slender, apex paddle shaped, truncated with irregular teeth on ventromesal margin. Connective nearly as wide as long (Figure 14D). Aedeagal shaft straight in lateral view, apical 1/3 tapering, bearing three pairs of retrorse processes, apical processes tiny, curved dorsally; ventral spines directed ventrally and tapering, moderately long and slightly divergent; lateral processes short, directed ventrally (Figure 14F). Gonopore slit-shaped, at middle of stem above caudal spines on ventral surface (Figure 14E).

Material examined. CHINA·1♂; Xinjiang Prov., Kanasi; 14 July 2016; D.Q. Ai leg.; NWAFU.

Distribution. Austria, Belgium, China (Xinjiang), Czech, Denmark, Estonia, Finland, France, Germany, Greece, Hungary, Italy, Kazakhstan, Kyrgyzstan, Latvia, Lithuania, Moldova, Mongolia, Netherlands, Norway, Poland, Romania, Russia, Slovakia, Sweden, Switzerland, Ukraine, United Kingdom, Uzbekistan, former Yugoslavia.

Diagnosis. It was originally named *trifasciatus*, but Koçak [95] discovered that this name was a homonym and proposed the replacement name *laevus*. Hamilton [7] suggested that *P. grisea* Mitjaev, the type of which has the styles fused to the aedeagus, is probably an abnormal form of *P. laevus*. *Aphrodes turkestanicus* described by Dubovsky [93] from Fergana Valley (Uzbekistan) is also a synonym of the species. Different individuals of *P. laevus* may have the base of the caudal processes of the aedeagus fused or separated [4,19].

**Figure 14 insects-14-00291-f014:**
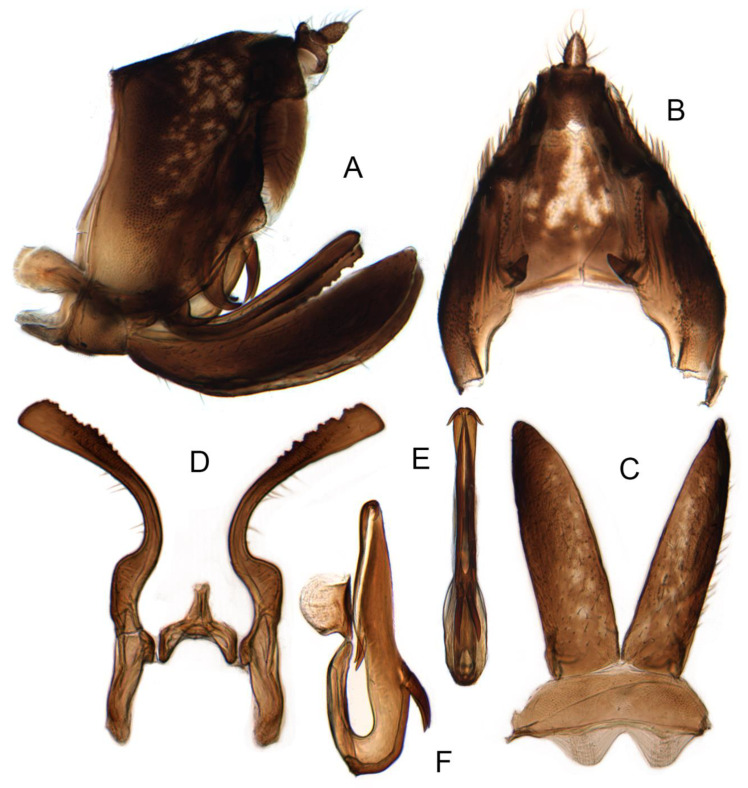
(**A**–**F**) *Planaphrodes laevus*. (**A**) pygofer, lateral view. (**B**) genital capsule, ventral view. (**C**) subgenital plates, ventral view. (**D**) styles and connective, ventral view. (**E**) aedeagus, ventral view. (**F**) aedeagus, lateral view.

#### 3.3.5. *Planaphrodes baoxingensis* Liang & Dai, sp. nov. (Figure 5C,G,K and Figure 15A–F)

Description. Length (including wings): male 5.1 mm.

Crown black, flattened and rugose. Ocelli red (Figure 5C). Face brown, with distinct dark brown mottling especially on genae, anteclypeus and base of frontoclypeus. Eyes black (Figure 5K). Pronotum black, with brown band on posterior margin. Scutellum dark. Forewings dark brown, opaque, veins indistinct, with narrow irregular yellowish transverse band situated in subapical region interrupted with dark brown and yellowish cloudy patch situated in basal 1/3, apex yellowish (Figure 5C). Abdomen dark brown (Figure 5G).

Pygofer well sclerotized, taller than long, caudal margin slightly concave, with distinct round posteroventral lobe covered with tiny setae and short hook-like process at middle of caudal submargin directed medially with preapical denticle (Figure 15A,B). Subgenital plates broad, curved, parallel-sided at basal half in ventral view, narrowed to rounded apex, with numerous irregularly arranged short setae (Figure 15C). Styles slender, with long crescent-shaped apical lobe denticulate on ventral margin. Connective short, Y-shaped (Figure 15D). Aedeagal shaft strongly compressed, in lateral view narrowed near base, widened near middle, gradually narrowing in its distal part, with four pairs of retrorse processes, apical processes tiny and hook-like; dorsal processes tapering apically and directed posteriorly, moderately long and slightly divergent; caudal processes tapering apically and directed ventrally; lateral processes small, directed ventrally (Figure 15E,F). Gonopore slit-shaped, on ventral surface near basal 1/3 above caudal spines (Figure 15E,F).

Material examined. Holotype, CHINA·1♂; Sichuan Prov., Baoxing, Fengtongzhai; 3 August 2004; X.J. Yang and H.R. Hua Leg.; NWAFU.

Etymology. The species name refers to the type locality.

Diagnosis. This new species is very similar to *P. nigritus* (Kirschbaum), which is widely distributed in the Palaearctic region but can be distinguished from the latter by the posteriorly directed dorsal spines and smaller lateral spines of the aedeagus (Figure 15F).

**Figure 15 insects-14-00291-f015:**
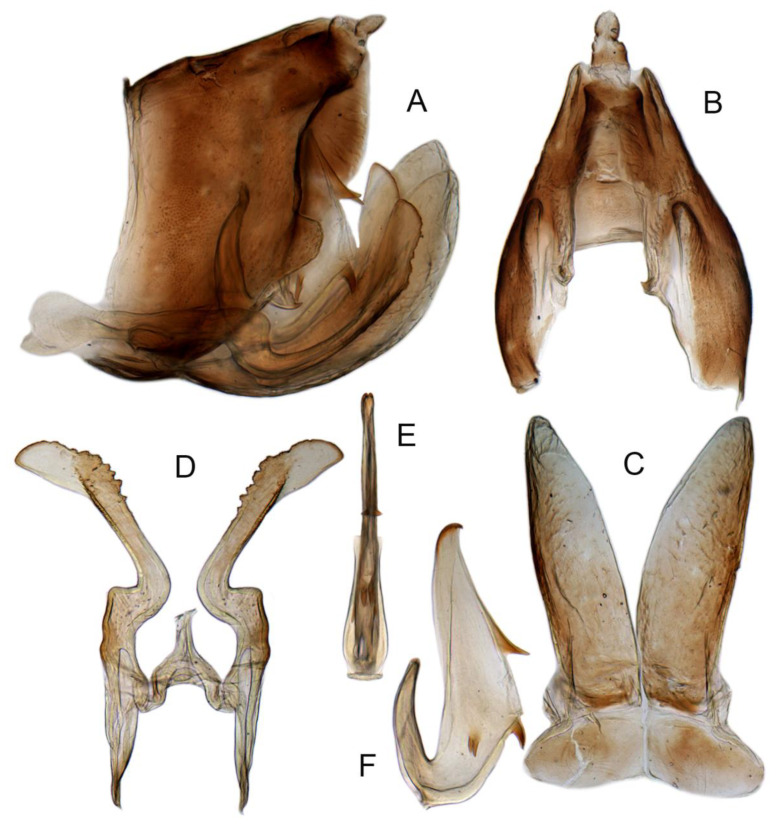
(**A**–**F**) *Planaphrodes baoxingensis* sp. nov. (**A**) pygofer, lateral view. (**B**) genital capsule, ventral view. (**C**) subgenital plates, ventral view. (**D**) stylets and connective, ventral view. (**E**) aedeagus, ventral view. (**F**) aedeagus, lateral view.

#### 3.3.6. *Planaphrodes faciems* Liang & Dai, sp. nov. (Figure 5D,H,L and Figure 16A–F)

Description. Length (including wings): male 5.0 mm.

Crown black, shagreened and upcurved, with some small yellow spots near anterior margin. Ocelli red (Figure 5D). Face brown, frontoclypeus with distinct dark brown mottling and oblique striations on both sides. Eyes black (Figure 5L). Pronotum black with indistinct dark brown mottling in posterior margin. Scutellum black, a shallow spot at apex. Forewings brown, opaque, veins indistinct, with a large irregular yellowish transverse band situated in subapical region and yellowish cloudy patch situated in basal 1/3 (Figure 5D). Abdomen brown. Legs brown (Figure 5H).

Pygofer well sclerotized, taller than long, caudal margin slightly concave, with distinct posteroventral lobe covered with tiny setae and hook-like process at middle of caudal submargin directed medially with a denticle (Figure 16A,B). Subgenital plates broad, curved, parallel-sided at basal half in ventral view, narrowed to rounded apex, with numerous irregularly arranged short setae (Figure 16C). Styles slender, with long crescent-shaped apical lobe denticulate on ventral margin. Connective short, Y-shaped (Figure 16D). Aedeagal shaft in caudal view strongly compressed(Figure 16E), in lateral view narrowed in its basal portion, widened in its middle part, gradually narrowing in its distal part, with five pairs of retrorse processes, apical processes tiny and directed caudally; dorsal processes triangular and directed caudally, moderately long and slightly divergent; caudal processes tapering apically and directed ventrally; between dorsal and caudal spines, with pair of small spines; lateral processes tiny and short, directed ventrally (Figure 16F). Gonopore slit-shaped, on ventral surface at basal 1/3 above caudal spines (Figure 16E,F).

Material examined. Holotype, CHINA·1 ♂; Hubei Prov., Wufeng, Houhe National Nature Reserve; 14 Aug. 2006; 1500 m; L. Lu leg.; NWAFU.

Etymology. This specific epithet is derived from the Latin word “faciem”, referring to the facelike color pattern of the forewings.

Diagnosis. This species is very similar to *P. nigricans*, *P. nigritus* and *P. faciems* but can be distinguished by the caudal spines directed caudally and presence of a pair of small spines between the caudal and ventral spines (Figure 16F).

**Figure 16 insects-14-00291-f016:**
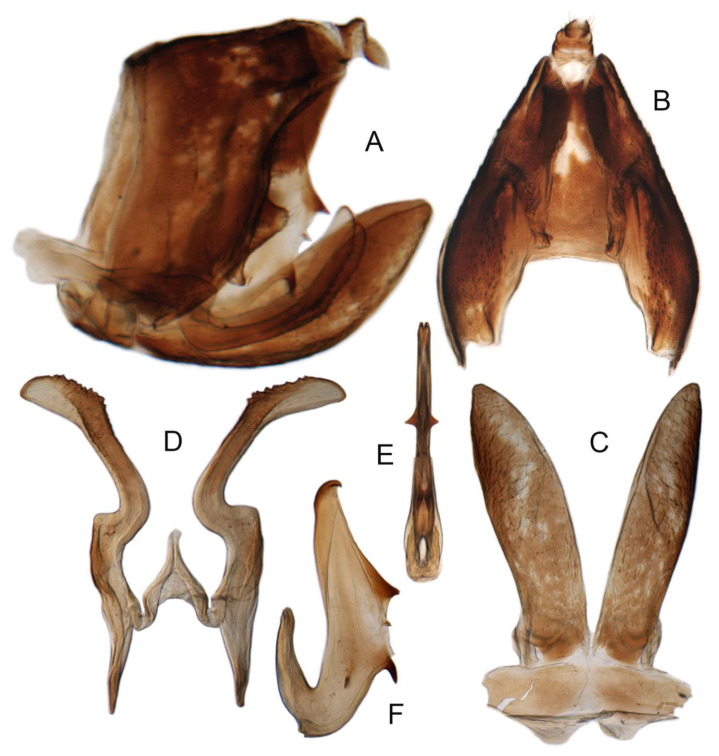
(**A**–**F**) *Planaphrodes faciems* sp. nov. (**A**) pygofer, anal tube and subgenital plate, lateral view. (**B**) genital capsule, ventral view. (**C**) subgenital plates, ventral view. (**D**) stylets and connective, ventral view. (**E**) aedeagus, ventral view. (**F**) aedeagus, lateral view.

## 4. Discussion

This study is the first attempt to elucidate relationships among genera of Aphrodini and within species of *Planaphrodes* using explicit phylogenetic methods. Our results support Hamilton’s recognition of four genera within the tribe and agree to some extent with his intuitive phylogeny based on aedeagal structure. Our phylogeny is difficult to compare to Hamilton’s because his tree diagram implies that various extant species are directly ancestral to others. Nevertheless, our results are consistent with Hamilton’s suggestion that *P. vallicola*, alone among *Planaphrodes* species, retains the ancestral position of the aedeagal spines, making it the most plesiomorphic species in the genus. Hamilton [7] also suggested that *P. nigricans* and *P. monticola* form a lineage characterized by having two pairs of spines close together on the caudal margin of the aedeagus and that this lineage is sister to the lineage, giving rise to most of the remaining species, characterized by a slender, parallel-margined aedeagal shaft. Within the latter group, *P. sahlbergii* and *P. angulaticeps* have the apical shaft spines projecting caudally in lateral aspect. These views are consistent with our phylogenetic results.

More broadly, our analysis provides morphological support for the sister-group relationship of *Planaphrodes* to *Aphrodes.* These genera share a strongly produced head, flat vertex between ocelli and sharp transition of vertex to face but differ in the coloration of the forewing (spotted or banded in *Planaphrodes* versus unmarked in *Aphrodes*) and the shape of the aedeagal shaft (strongly compressed in *Planaphrodes* but cylindrical in *Aphrodes*). The other two genera of Aphrodini are also supported as monophyletic by our analysis, with *Stroggylocephalus* supported by two unique synapomorphies (crown parallel-margined; flat ellipse gonopore shape) and *Anoscopus* supported by one homoplasious character change (aedeagus with apical spines oriented anteroventrad).

Our morphology-based phylogenetic estimate shows some large-scale biogeographic structure within *Planaphrodes* and suggests that regional faunas in Europe and East Asia may have arisen from widespread Eurasian ancestors. For example, Clade 2 includes a subclade containing three species (*P. nigrigans*, *P. faciems* and *P. baoxingensis*) restricted to East Asia that is successively sister to *P. nigritus*, *P. monticola* and *P. bifasciatus*; these three widespread species occur across Eurasia. Similarly, Clade 1 includes widespread species *P. laevus* that is sister to a clade comprising species with more restricted distributions in Europe or Asia. *Paraphrodesvallicola*, which is sister to the clade comprising the remaining species of the genus, occurs in Kazakhstan and adjacent parts of southern Russia, supporting a possible Central Asian origin for the genus. *Planaphrodes* is the only genus of Aphrodini well represented in East Asia. Other genera of Aphrodini are largely or entirely restricted to Europe and western Asia, with a few species apperantly adventive in the Nearctic region.

Some changes to the taxonomy of *Planaphrodes* suggested by our results will need to be confirmed through further study. Based on published descriptions and figures [17,31], *P. iranicus* and *P. nisamiana* are similar in appearance and male genitalia and occur in neighboring countries (Azerbaijan and Iran). Although they differ in three characters included in our matrix pertaining to small details of aedeagal structure (Figure 2), their overall morphological similarity suggests that they may be synonyms. This should be confirmed through study of the types and other specimens from those countries and analysis of molecular data. *P. laevus* and *P. lusitanicus* differ in only one character scored in our matrix (Figure 2), so we also suspect these two taxa may be synonyms. The former is recorded from several Eurasian countries, while the latter has been reported only from Portugal. *P. modicus* also resembles *P. laevus* in aedeagal structure (with base of caudal aedeagal processes fused) except for the poorly developed caudal processes. This species was described by Logvinenko [28] based on a single male specimen from Moldavia and there have not been further reports of this species in the literature. Additional study is needed to confirm whether the type of *P. modicus* is a malformed individual of *P. laevus*. The relationships within subclade ((*P. sahlbergii* + *P. angulaticeps*) + (*P. araxicus* + *P. modicus*)) are not consistently resolved among equally parsimonious trees.

Due to the limited numbers of specimens available, we were not able to test the validity of morphology-based species concepts in *Planaphrodes*, so further studies are needed to address this issue. Although the recent acoustic studies of Tisheshkin [21] have provided some corroboration of morphology-based species concepts currently used in *Planaphrodes*, further collecting throughout the known range of the genus and additional comparative morphological study are needed to elucidate the extent of morphological variation in species currently known from a few or single individuals. Ultimately, molecular phylogenetic and phylogeographic studies as well as acoustic studies of species not recorded by Tishechkin [21] will be needed to validate current species concepts.

## Figures and Tables

**Figure 1 insects-14-00291-f001:**
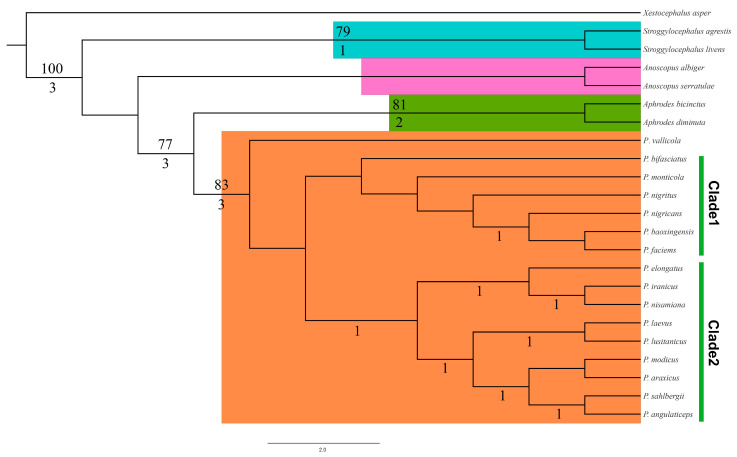
One of 4 most parsimonious cladograms of the tribe Aphrodini (Blue—*Stroggylocephalus*; Pink—*Anoscopus*; Green—*Aphrodes*; Orange—*Planaphrodes*) phylogenetic relationships indicated with bootstrap values (above branches) and Bremer’s decayindices (below branches).

**Figure 2 insects-14-00291-f002:**
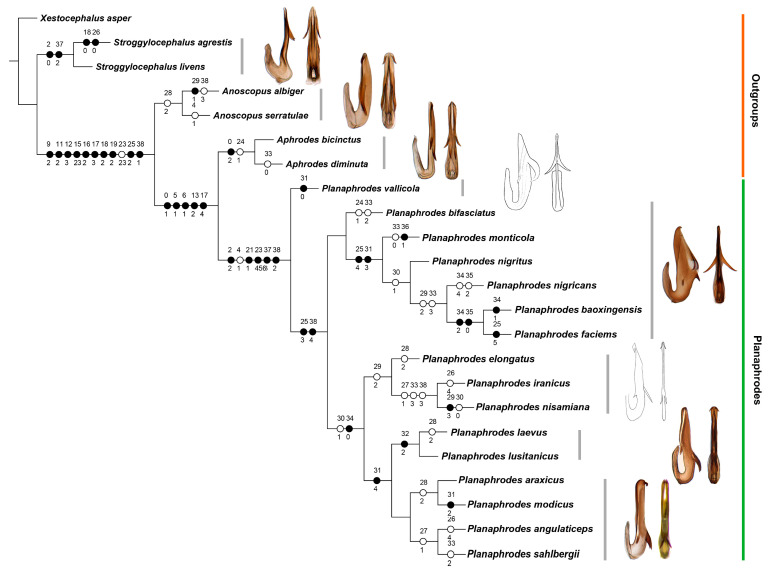
Phylogenetic hypothesis for the tribe Aphrodini from Figure 1 showing character state changes. Numbers above the circles refer to characters and those below refer to character state. Filled and open circles represent synapomorphies and homoplasious character changes, respectively. Lateral and ventral views of aedeagus from relative specimen have also been provided.

**Figure 3 insects-14-00291-f003:**
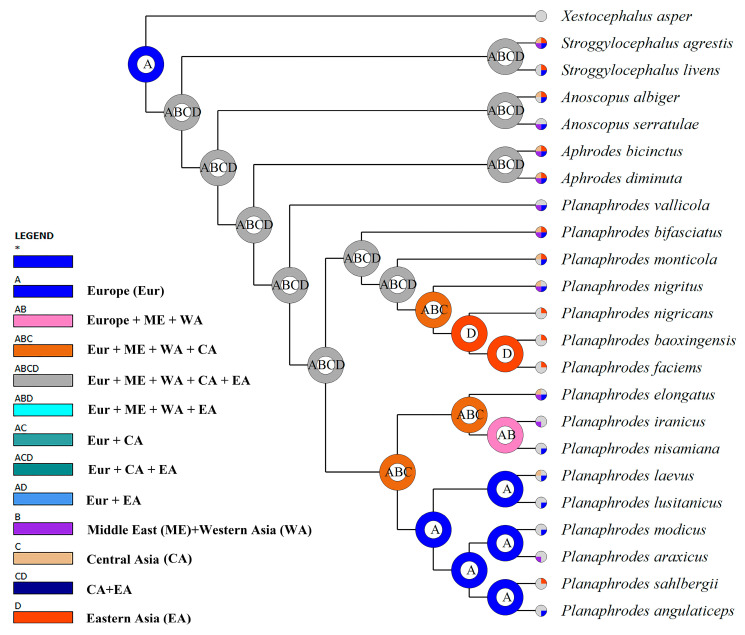
Bayesian binary MCMC reconstruction of biogeographic associations using the consensus of the four original MP trees by TNT.

**Table 1 insects-14-00291-t001:** Character state distribution used in cladistic analysis.

Species	0	1	2	3	4	5	6	7	8	9	10	11	12	13	14	15	16	17	18	19	20	21	22	23	24	25	26	27	28	29	30	31	32	33	34	35	36	37	38
*Xestocephalus asper*	0	0	1	0	0	0	0	0	0	0	0	0	0	0	0	0	0	0	1	0	0	0	0	0	0	0	-	-	-	-	-	-	-	-	-	-	-	0	0
*Aphrodes bicinctus*	2	1	1	1	0	1	1	1	1	2	1	2	3	2	1	2	2	4	2	2	1	0	1	3	1	2	-	-	-	0	0	1	0	1	-	-	-	1	1
*Aphrodes diminuta*	2	1	1	1	0	1	1	1	1	2	1	2	3	2	1	2	2	4	2	2	1	0	1	3	1	2	-	-	-	0	0	1	0	0	-	-	-	1	1
*Anoscopus albiger*	0	1	1	0	0	0	0	0	1	2	1	2	3	1	1	3	3	3	2	2	1	0	1	2	0	2	2	0	2	1	2	-	-	-	-	-	-	1	3
*Anoscopus serratulae*	0	1	1	0	1	0	0	0	?	?	?	?	?	?	?	?	?	?	?	2	1	0	1	3	0	2	2	0	2	0	2	-	-	-	-	-	-	1	1
*Stroggylocephalus agrestis*	0	1	0	1	0	0	0	1	1	1	1	1	1	1	1	1	1	1	0	1	1	0	1	1	0	1	0	0	0	-	-	-	-	-	-	-	-	2	0
*Stroggylocephalus livens*	0	1	0	1	0	0	0	1	1	1	1	1	2	1	1	1	1	2	1	1	1	0	1	1	0	1	1	0	1	-	-	-	-	-	-	-	-	2	0
*Planaphrodes angulaticeps*	1	1	2	1	?	?	?	?	?	?	?	?	?	?	?	?	?	?	?	?	1	1	1	5	0	3	4	1	0	0	1	4	1	0	0	-	-	3	4
*Planaphrodes araxicus*	1	1	2	1	1	1	?	?	?	?	?	?	?	?	?	?	?	?	?	?	1	1	1	0	0	3	3	0	2	0	1	4	1	0	0	-	-	3	4
*Planaphrodes baoxingensis*	1	1	2	1	1	1	1	1	1	2	1	2	3	2	1	3	3	5	2	2	1	1	1	6	0	4	3	1	0	2	1	3	1	3	1	0	0	3	4
*Planaphrodes bifasciatus*	1	1	2	1	1	1	1	1	?	?	?	?	?	?	?	?	?	?	?	?	1	1	1	6	1	3	-	-	-	0	0	1	1	2	3	1	0	3	4
*Planaphrodes elongatus*	1	1	2	1	?	?	1	?	?	?	?	?	?	?	?	?	?	?	?	?	1	1	1	4	0	3	3	0	2	2	1	1	1	1	0	-	-	3	4
*Planaphrodes faciems*	1	1	2	1	1	1	1	1	1	2	1	2	3	2	1	3	3	5	2	2	1	1	1	6	0	5	3	1	0	2	1	3	1	3	2	0	0	3	4
*Planaphrodes iranicus*	1	1	2	1	?	?	1	?	?	?	?	?	?	?	?	?	?	?	?	?	1	1	1	4	0	3	4	1	0	2	1	1	1	3	0	-	-	3	3
*Planaphrodes laevus*	1	1	2	1	1	1	1	1	1	2	1	2	3	2	1	3	?	?	2	2	1	1	1	5	0	3	3	0	2	0	1	4	2	1	0	-	-	3	4
*Planaphrodes lusitanicus*	1	1	2	1	1	1	1	?	?	?	?	?	?	?	?	?	2	4	2	?	1	1	1	5	0	3	3	0	0	0	1	4	2	1	0	-	-	3	4
*Planaphrodes modicus*	1	1	2	1	1	1	1	1	?	?	?	?	3	?	?	?	2	4	2	?	1	1	1	0	0	3	3	0	2	0	1	2	1	0	0	-	-	3	4
*Planaphrodes monticola*	1	1	2	1	1	1	1	1	1	2	1	2	3	2	1	3	3	5	2	2	1	1	1	6	0	4	3	1	0	0	0	3	1	0	3	1	1	3	4
*Planaphrodes nigricans*	1	1	2	1	1	1	1	1	1	2	1	2	3	2	1	3	3	5	2	2	1	1	1	6	0	4	3	1	0	2	1	3	1	3	4	2	0	3	4
*Planaphrodes nigritus*	1	1	2	1	1	1	1	1	?	?	?	?	?	?	?	?	?	?	?	?	1	1	1	6	0	4	3	1	0	0	1	3	1	1	3	1	0	3	4
*Planaphrodes nisamiana*	1	1	2	1	1	?	1	?	?	?	?	?	?	?	?	?	2	4	2	?	1	1	1	4	0	3	3	1	0	3	0	1	1	3	0	-	-	3	3
*Planaphrodes sahlbergii*	1	1	2	1	1	1	1	1	1	2	1	2	3	2	1	3	2	4	2	2	1	1	1	0	0	3	3	1	0	0	1	4	1	2	0	-	0	3	4
*Planaphrodes vallicola*	1	1	2	1	1	1	1	?	?	?	?	?	3	?	?	?	2	4	2	?	1	1	1	6	0	2	-	-	-	0	0	0	-	-	4	2	-	3	2

**Table 2 insects-14-00291-t002:** Comparative morphology of Aedeagus and style of *Planaphrodes* (the styles of some species were not provided in the references).

Species Name	Aedeagus, Lateral View	Aedeagus, Caudal View	Style	Species Name	Aedeagus, Lateral View	Aedeagus, Caudal Views	Style
*angulaticeps*	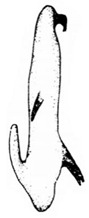	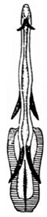	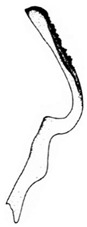	*araxicus*	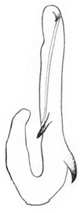	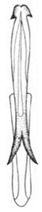	
*baoxingensis*	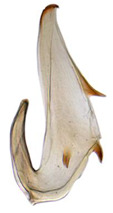	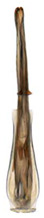	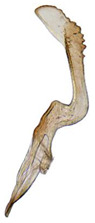	*bifasciatus*	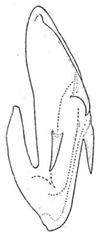	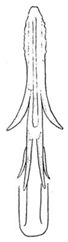	
*elongatus*	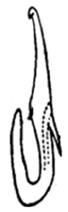	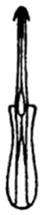		*faciems*	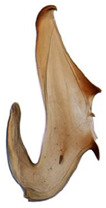	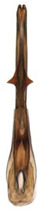	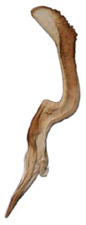
*iranicus*	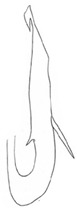	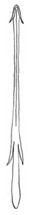		*laevus*	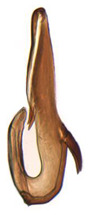	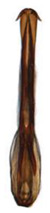	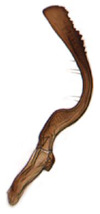
*lusitanicus*	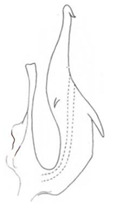	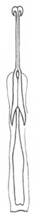	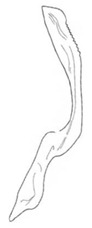	*modicus*	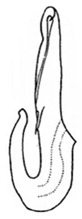	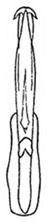	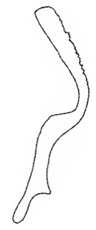
*monticola*	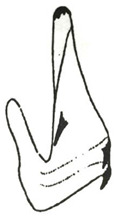	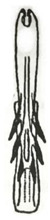	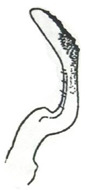	*nigricans*	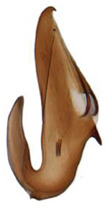	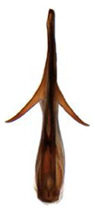	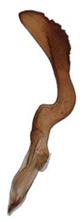
*nigritus*	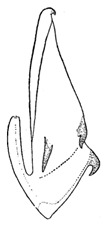	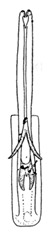		*nisamiana*	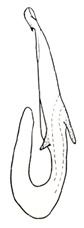	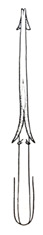	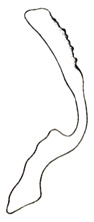
*sahlbergii*	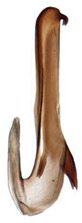	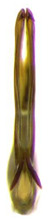	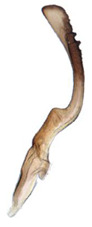	*vallicola*	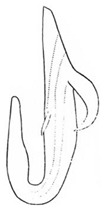	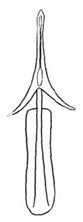	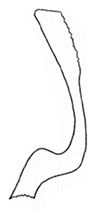

Note: refer to Table 2 for species figures, *P. araxicus* from Logvinenko, 1971 [29] (Figure 1: 1 and 2); *P. angulaticeps* from Cantoreanu, 1968 [18] (*P. dobrigicus* syn. Figure 1: A–C); *P. bifasciatus* from Ribaut, 1952 [35] (Figures 909–910); *P. elongatus* from Emeljanov, 1964 [19] (Figure 173: 23–24); *P. iranicus* from Dlabola, 1971 [31] (Figures 16 and 17); *P. lusitanicus* from Rodrigues, 1968 [30] (Figure 3: A–C); *P. modicus* from Logvinenko, 1966 [28] (Figures 43–45); *P. monticola* from Anufriev et al., 1988 [9] (Figure 113: 11–16); *P. nigritus* from Ribaut, 1952 [36] (Figures 906–907); *P. nisamiana* from Logvinenko, 1983 [17] (Figure 3: 5–7); *P. vallicola* from Mitjaev, 1979 [32] (Figures 6, 9 and 10).

## Data Availability

All data are contained within the article.

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
