# Peer review of "Phylogenetic Analysis of the Genus Planaphrodes Hamilton (Hemiptera, Cicadellidae, Aphrodinae) Based on Morphological Characteristics, with Revision of Species from China, Korea and Japan"

_insects, 2023, doi:10.3390/insects14030291_

Round 1

Reviewer 1 Report

The authors conducted the morphology-based phylogeny of the genus Planaphrodes for the first time, and provided the checklist and key to the Planaphrodes species. Also, two new synonyms were proposed and two new species were described. It’s a valuable work, but there are still some issues need to be clarified.

Please check carefully and italicize all genus and species names. (e.g., line 21, line 438, line 473, etc.).

Please unify the expression of phylogenetic relationships among species/genus. In line 22, the authors used (Stroggylocephalus (Anoscopus (Planaphrodes, Aphrodes))) to express the phylogenetic relationship among these four genera, but in line 305, (P. sahlbergii + P. angulaticeps + (P. araxicus + P. modicus)) was used, etc.

From 2.2 section, I got the information that not all the specimens of Planaphrodes were available, only 6 ingroup and 7 outgroup species distributed in China were examined, but in table 2, the aedeagus and style of 15 Planaphrodes species are provided. I think the origin of figures of the 9 unexamined should be clearly stated (redrew according to literature or cited).

From the whole paper, I didn’t find the words to explain the reason and characteristics of proposing the new synonyms, please supplement.

Line 45 It seems the sentence should be “In Hamilton’s comprehensive review of the Northern Hemisphere Aphrodina, he erected Planaphrodes with the type species Acucephalus tricincta (Curtis) and included fifteen species based on his examination of a series of type material for species in the genus”. Please check it carefully.

Line 275: It seems there is an extra "and".

Line 302: According to figure 1, the interspecific relationship should be ((P. laevus + P. lusitanicus) + ((P. sahlbergii + P. angulaticeps) + (P. araxicus + P. modicus))), please correct. This also applies to line 305 and 882.

Line 323: change “are” to “is”.

Line 524: From the letters marked on the provided figures, “Figure 4. A-H” should be change to “Figure 4. A-I”.

Line 699: Although it has been stated in the main text that the male genitalia of Planaphrodes bifasciatus is reproduced from previous study, the notes after the title of figure 12 are also needed.

Both Xizang and Tibet refer to the same one autonomous region of China. I think one should be used in a whole article, not both. Please unify.

Reviewer 2 Report

This is a comprehensive review of this leafhopper genus.  It is very well presented and illustrated.

Just one comment about abundance of some species. They can be very commonly found in certain situations with vacuum sampling. I assume that they live close to the ground and may not be found so effectively with a sweep net. 

Author Response

Just one comment about abundance of some species. They can be very commonly found in certain situations with vacuum sampling. I assume that they live close to the ground and may not be found so effectively with a sweep net. 

Thanks for your nice advice. We will try to collect the specimens close to the ground with vacuum sampling in the future.

Reviewer 3 Report

The manuscript Phylogenetic analysis of the genus Planaphrodes Hamilton  (Hemiptera, Cicadellidae, Aphrodinae) based on morphological characteristics, with revision of species from China, Korea and Japan. Treats about the morphology-based phylogeny of the Holarctic leafhopper genus Planaphrodes Hamilton and is reconstructed for the first time, based on 39 discrete male adult morphological characters. The fauna of Planaphrodes from China, Japan and Korea are reviewed and six species are recognized, including two new species and synonymies are included. A checklist and a key to species of Planaphrodes are provided.

Leafhoppers of the  tribe are important because they are common on herbaceous plants, usually in meadows and pastures, and some species live and feed on roots beneath the surface litter. Some species also breed on legumi nous crops, e.g., alfalfa.

This contribution is important but it can not be published in this way. Major corrections should be done.

There is information  to be included, for example mesaurements, maps with geographic distribution, and many editorial changes to be performed. Please see the corrections done in the manuscript. The drawings should be homogeneous. References are not done in the same way.

Include a first protograph showing the characters used with the letters abbreviated of that character. Tables with measurements ang geographic distribution maps.

A very important contribution but should be redone.

Round 2

Reviewer 1 Report

The authors have corrected and responded to all the comments I suggested. But I found that some of the photographed figures don't contain scale bars, please supplement.

Author Response

The authors have corrected and responded to all the comments I suggested. But I found that some of the photographed figures don't contain scale bars, please supplement.

Response: Thanks for your advice. In this manuscript, all photos of habitus and male genitilia are sufficient to show diagnosis characters of described species. We have also given detailed descriptions including the length of described species. We think that is not essential to supplement the scale bars. Moreover, the photographed figures without scale bars including Figures 4A-I, 8A,B, E-I, were taken based on the specimens borrowed from South Korea and Japan (some are type specimens). Unfortunately, these specimens have been returned. We do not believe that the scale bars will have an impact on our results.